



# Measurement report: Regional characteristics of seasonal and long-term variations in greenhouse gases at Nainital, India and Comilla, Bangladesh

Shohei Nomura[1], Manish Naja[2], Md. Kawser Ahmed[3], Hitoshi Mukai[1], Yukio Terao[1], Toshinobu Machida[1], Motoki Sasakawa[1], Prabir K. Patra[4]

[1]Center for Global Environmental Research, National Institute for Environmental Studies, 16-2 Onogawa, Tsukuba, Ibaraki, 305-8506, Japan
[2]Aryabhatta Research Institute of Observational Sciences, Manora Peak, Nainital Uttarakhand 263129, India
[3]Department of Oceanography, Faculty of Earth & Environmental Sciences, University of Dhaka, Dhaka-1000, Bangladesh
[4]Research Institute for Global Change, JAMSTEC, 3173-25 Showa-machi, Yokohama, 236-0001, Japan

*Correspondence to*: Shohei Nomura (nomura.shohei@nies.go.jp) Tel.: +81 298502370; fax: +81 298502960

**Abstract**

Emissions of greenhouse gases (GHGs) from the Indian subcontinent have increased during the last 20 years along with rapid economic growth, however, there remains a paucity of GHG measurements for policy relevant research. In northern India and Bangladesh, agricultural activities are considered to play an important role on GHGs concentrations in the atmosphere. We performed weekly air sampling at Nainital (NTL) in northern India and Comilla (CLA) in Bangladesh from 2006 and 2012, respectively. Air samples were analyzed for dry-air gas mole fractions of $CO_2$, $CH_4$, $CO$, $H_2$, $N_2O$, and $SF_6$, and carbon and oxygen isotopic ratios of $CO_2$ ($\delta^{13}C$-$CO_2$ and $\delta^{18}O$-$CO_2$). Regional characteristics of these components over the Indo-Gangetic Plain are discussed compared to data from other Indian sites and Mauna Loa, Hawaii (MLO), which is representative of marine background air.

We found that the $CO_2$ mole fraction at both NTL and CLA had two seasonal minima in February–March and September, corresponding to crop cultivation activities that depend on regional climatic conditions. The carbon isotopic signature also suggested that photosynthetic $CO_2$ absorption by crops cultivated in each season contributes differently to lower $CO_2$ mole fractions. The $CH_4$ mole fraction of NTL and CLA in August–October showed high values (i.e., sometimes over 4,000 ppb at CLA) due to the influence of $CH_4$ emissions from the paddy fields in addition to the other sources due to the hot and humid climatic conditions. High $CH_4$ mole fractions sustained over months at CLA were a characteristic feature in the Indo-Gangetic Plain. The $CO$ mole fractions at NTL were also high and showed peaks in May and October, while CLA had much higher peaks in October–March due to the influence of human activities such as emissions from biomass burning and brick production. The $N_2O$ mole fractions at NTL and CLA increased in June–August and November–February, which coincided with the application of nitrogen fertilizer and the burning of biomass such as the harvest residues and dung for domestic cooking. Based on $H_2$ seasonal variation at both sites, it appeared that the emissions in this region were related to biomass burning in addition to production from the reaction of OH and $CH_4$. The $SF_6$ mole fraction was similar to that at MLO, suggesting that there were few anthropogenic emission sources in the district.

The variability of $CO_2$ growth rate at NTL was different from the variability in the $CO_2$ growth rate at MLO, which is more closely linked with the El Niño Southern Oscillation (ENSO). In addition, the growth rates of the $CH_4$ and $SF_6$ mole fractions at NTL showed an anticorrelation with those at MLO, indicating that the frequency of southerly air masses strongly influenced these mole fractions. These finding showed that rather large regional climatic conditions considerably controlled interannual variations in GHGs, $\delta^{13}C$-$CO_2$, and $\delta^{18}O$-$CO_2$ through changes in precipitation and air mass.





**Keywords**
Northern India, Bangladesh, Greenhouse gases variation, Isotope ratio of $CO_2$, Local emissions
**1 Introduction**

The atmospheric mole fractions of $CO_2$, $CH_4$, $N_2O$ and many other greenhouse gases (GHGs) are increasing until the

recent years globally. As for $CO_2$, rapid increases in $CO_2$ emissions from emerging countries contribute strongly to acceleration
of the growth rate of its mole fraction (Friedlingstein et al., 2019). For instance, anthropogenic $CO_2$ emission of India has
increased in 2017 it reached to 2.45 $GtCO_2$ $yr^{-1}$ which was the third highest in the world (Muntean et al., 2018). Therefore,
South Asian region must be important to evaluate GHG in the future. Patra et al. (2013) calculated the $CO_2$ flux in South Asia
using top-down and bottom-up methods and reported that $CO_2$ fluxes in top-down and bottom-up were $-104 \pm 150$ $TgCyr^{-1}$
and $-191 \pm 193$ $TgCyr^{-1}$. In other words, $CO_2$ was absorbed in South Asia, however, the error of $CO_2$ flux was very large
because there are few measured GHG moles fractions in the South Asian region.

Several observations on GHGs mole fractions in the atmosphere have been done around India. The first systematic

monitoring for GHGs mole fractions and carbon isotopic ratio in the South Asian region was performed by Bhattacharya et al.
(2009). They carried out monitoring at Cape Rama station (CRI) (15.1°N, 73.9°E, 60 m a.s.l.) on the West Coast of India from
1993 and found that (1) the $CH_4$ and CO mole fractions increased in October–March when the air mass came from the northeast
(inland), and decreased in June–August when the air mass came from southwest (ocean); (2) the $CO_2$, $CH_4$, CO, $H_2$ and $N_2O$
mole fractions in June–August were generally at the same levels at the background sites at the observatory in Seychelles Island
and Hawaii Island; (3) the seasonal cycle and phase in $CH_4$ and CO mole fractions were quite similar and their correlation
coefficient was high, generally because they originated from anthropogenic emissions in India. Therefore, it became clear that
GHG mole fractions are greatly changed by the seasonal wind and that the Indian subcontinent has strong $CH_4$ and CO
emissions (Patra et al., 2009).

In recent decades, a few more research groups have commenced flask sampling or continuous GHG measurements in

India. Sharma et al. (2013) measured atmospheric $CO_2$ mole fractions at Dehradun in northern India in 2009 and detected that
the $CO_2$ mole fraction decreased twice a year (March and September) due to vegetation activity. Ganesan et al. (2013) measured
the $CH_4$, $N_2O$, and $SF_6$ mole fractions in December 2011 to February 2013 at Darjeeling in northeastern India and found that
(1) $CH_4$ mole fractions had a positive correlation with the $N_2O$ mole fraction, and that those mole fractions increased due to
emissions from anthropogenic activities when air masses came from the Indo-Gangetic Plain; (2) $SF_6$ emissions in the region
showed a weak signal. Chandra et al. (2016) measured the $CO_2$ and CO mole fractions at Ahmedabad in western India and
detected a decrease in the mole fraction when the air mass comes from southwest (ocean) and an increase in the mole fraction
when the air mass comes from northeast (inland). Tiwari et al. (2014) analyzed the spatial variability of atmospheric $CO_2$ mole
fractions using models over the Indian subcontinent and began the flask sampling at Sinhagad in western Ghats. They showed
(1) the seasonal variation of the $CO_2$ mole fraction in southern India differed with the variation on the Indo-Gangetic Plain in
northern India due to the differences in air mass transportation and anthropogenic activity; (2) the $CO_2$ mole fraction in July–
October at Sinhagad was lower than the mole fraction of CRI on the west coast India because of the influence of photosynthesis
by the regional forest ecosystem.

Sreenivas et al. (2016) measured the mole fractions of $CO_2$ and $CH_4$ at Shadnagar in central India and reported that the

$CO_2$ and $CH_4$ mole fractions were strongly positively correlated to anthropogenic sources. Lin et al. (2015) commenced the
most ambitious flask sampling network, with sites at Pondicherry (PON) on the southeast coast India, Port Blair (PBL) on
Andaman Island, and Hanle (HLE) in northwestern Himalaya. They reported that (1) the mole fractions of $CH_4$, CO, and $N_2O$
at PON and PBL were relatively high in comparison with those at HLE; (2) seasonal variations in GHGs at PON and PBL
were quite different from the variation at HLE because the former two sites were exposed to the influence of air masses





originating from areas of anthropogenic activities. In addition to these studies at ground sites, recently aircraft-base
observations over Indo-Gangetic Plain such as CONTRAIL have been also carried out actively, evaluating seasonal variation
of CO2 mole fraction (Umezawa et al, 2016).
Thus, the GHG observation program in Indian region is expanding gradually, however, observation sites and
characterization of GHG behavior and their long-term trends remain limited. In this work, we present an analysis of long record
(14 years) of various GHGs mole fraction and isotopic ratios of $CO_2$ ($\delta^{13}C$-$CO_2$ and $\delta^{18}O$-$CO_2$) at Nainital, India on a mountain
site near the Himalayan mountain range, which can be considered as a background site representing Northern Indian air, and
which is partly influenced by anthropogenic activities from the Indo-Gangetic Plain. We also show a similar 8-year GHG
record at Comilla, Bangladesh located in the eastern edge of Indo-Gangetic Plain, where agricultural activities are believed to
the main factors for GHG emissions. The levels and seasonal variabilities of GHGs mole fraction at these sites are discussed
compared to those at other Indian sites reported previously, along with the local precipitation and 72 hr back trajectory to
summarize the behavior of GHGs in this region. Relationship of mole fractions among GHGs are evaluated. We also describe
isotopic characteristics of $CO_2$ to consider contribution on absorption by $C_3$ and $C_4$ plants in each region.  Furthermore, we
analyze the relationships between the interannual variabilities in GHG growth rates and regional climatic condition such as the
Indian Dipole Mode Index (DMI) and the El Niño Southern Oscillation (ENSO) index.

## 2 Methods

### 2.1 Location

Figure 1 shows the locations of Nainital station (NTL) and Comilla station (CLA) where we performed weekly sampling.
The GHG observation sites in previous studies on the Indian subcontinent are also marked.
NTL is located at Aryabhatta Research Institute of Observational Sciences (ARIES) (29.36°N, 79.46°E, 1940 m a.s.l.)
on the top of Mt. Mauna Peak on the foot of the Himalaya mountain range facing the Indo-Gangetic Plain. Also, NTL is located
3 km south of Nainital city and no local residential building within 2 km from the station. Predominant wind direction at NTL
show the west-northwest during winter and east during summer (Naja et al., 2016), which mean that NTL might be influenced
mainly by the air mass passing through the Indo-Gangetic Plain. We estimated that the air of NTL is not strongly influenced
by local GHGs emissions nearby.
CLA is located at Comilla weather station of Bangladesh Meteorological Department (BMD) (23.43°N, 91.18°E, 30 m
a.s.l.) on the edge of farming village with a flat landscape in central Bangladesh. The surrounding area of CLA cover the paddy
fields and a few farmhouses. Farmers in Comilla burn the biomass (e.g. harvest residuals, firewood and dung) on daily basis
and it was expected that $CO_2$, $CH_4$, CO, $H_2$, $N_2O$ were emitted by the burning. CLA is considered to capture mainly the effect
of emission and sink in rural areas in eastern Indo-Gangetic Plain but may also capture some effect from nearby emissions.

### 2.2 Air sampling

Flask samples were collected from September 2006 in NTL and from June 2012 in CLA. Inlets were mounted at 7 m
above ground level (on the roof of the second floor of the station) in NTL and 8 m above ground level (on top of the 5 m tower
on the roof of the one-storey weather station building) in CLA. Air samples were collected in 1.5-L Pyrex flask through the
sampling line (Fig. 2[a]) at 2 p.m. (local time) once a week (usually on Wednesday). The sampling line contained a diaphragm
pump (MOA-P108-HB, GAST Co., Ltd.) and a freezer (VA-120, Taitec Co., Ltd.) for dehumidification by a glass trap. The
sampling flow rate was approximately 2 L min$^{-1}$ and the sample was passed through a -30 °C cooler and pressurized to 0.25
MPa after 10 min flushing through the sampling tube and flask. The sampled flasks were packed in a cardboard box and
transported to the laboratory of the Center for Global Environmental Research (CGER), National Institute for Environmental
studies, Japan (NIES) (transportation period: 3–7 days) for analyses.



## 2.3 Measurement methods

Air sample was passed through a -80 °C cold trap for dehumidification and was delivered to each instrument with a flow rate of 40 ml/min (see the analysis line in Fig. 2[b]). A nondispersive infrared analyser (NDIR; LI-COR, LI-6252) was used for $CO_2$ analysis, a gas chromatograph equipped with a flame ionization detector (GC-FID; Agilent Technologies, HP-5890 or HP-7890) was used to analyze $CH_4$, a gas chromatograph with a reduction gas detector (GC-RGD; Agilent Technologies, HP-5890+Trace Analytical RGD-2 or Peak Laboratories, Peak Performer 1 RCP) was used for CO and $H_2$ analyses, and a gas chromatograph with an electron capture detector or a micro electron capture detector (GC-ECD or GC-micro-ECD; Agilent Technologies, HP-6890) was used to analyze $N_2O$ and $SF_6$.

Dry-air mole fractions were measured against each of their working standard gases which were calibrated with NIES secondary standard gas series ($CO_2$-NIES09 scale, $CH_4$-NIES94 scale, CO-NIES09 scale, $H_2$-NIES96 scale, $N_2O$-NIES01 scale, and $SF_6$-NIES01 scale). Comparison between those scales and the National Oceanic and Atmospheric Administration (NOAA) scale in the 6[th] Round Robin intercomparison (NOAA/ESRL, 2019a) showed -0.04 to -0.09 ppm for $CO_2$, 3.7 to 4.1 ppb for $CH_4$, 4.0 to 4.4 ppb for CO, -0.61 to -0.69 for $N_2O$, and -0.03 to -0.06 ppt for $SF_6$. We evaluated that the NIES scales were almost the same as NOAA scales except for $CH_4$ which showed a bias that was beyond the measurement precision of our instrument.

After the mole fraction analysis, we used the remaining air inside the flask for analysis of $\delta^{13}C$-$CO_2$ and $\delta^{18}O$-$CO_2$. The air was introduced into two traps sequentially (-100 °C and -197 °C), which trapped $H_2O$ and $CO_2$, respectively. Finally, $CO_2$ was sealed in a glass tube. Air $\delta^{13}C$-$CO_2$ and $\delta^{18}O$-$CO_2$ were measured using the working standard $CO_2$ gas which was prepared in our laboratory by MT-252. The method for producing the working standard gas is similar to the method for producing the NIES Atmospheric Reference $CO_2$ for Isotopic Studies (NARCIS), which is used for interlaboratory-scale comparison (Mukai, 2001). The working standard scales of $\delta^{13}C$-$CO_2$ and $\delta^{18}O$-$CO_2$ are the same as those of NARCIS, which were measured by various institutions related to the World Meteorological Organization (WMO) (Mukai, 2003). The differences between NIES scales and INSTAAR (Institute of Arctic and Alpine Research) scales were 0.013–0.039‰ in the mean value range of -8.683 to -8.759‰ for $\delta^{13}C$-$CO_2$, and -0.017–0.022‰ in the mean value range of -1.956 to -9.299‰ of $\delta^{18}O$-$CO_2$ in the 6[th] Round Robin intercomparison (NOAA/ESRL, 2019a). The $\delta^{18}O$-$CO_2$ for atmospheric $CO_2$ in this study is expressed against the value of $CO_2$ evolved from VPDB calcite (i.e., VPDB-$CO_2$ scale, [IAEA, 1993, Brand et al., 2010]). Although the VSMOW scale is often used for $\delta^{18}O$ values of water, $CO_2$ evolved from VPDB calcite (VPDB-$CO_2$ scale) has similar $\delta^{18}O$ values of $CO_2$ equilibrated with VSMOW water, which is the reference gas of the VSMOW scale. The difference between them is only 0.263‰ (IAEA, 1993, Kim et al., 2015). Additionally, corrections for $N_2O$ bias and $\delta^{17}O$-$CO_2$ showed by Brand et al. (2010) were made to obtain final isotope ratios.

## 2.4 Reference dataset

For comparison with the data of NTL and CLA, we obtained weekly data ($CO_2$, $CH_4$, CO, $N_2O$, $SF_6$, $\delta^{13}C$-$CO_2$, and $\delta^{18}O$-$CO_2$) from the Mauna Loa Observatory (MLO) (19.54°N, 155.58°W, 3397 m a.s.l.) on the NOAA/ESRL website (NOAA/ESRL, 2019b). We also used biweekly data for $CO_2$, $CH_4$, CO, $H_2$, $N_2O$, and $\delta^{13}C$-$CO_2$ from Cape Rama, India (CRI) (15.08°N, 73.83°W, 60 m a.s.l.) on the website of World Data Centre for Greenhouse Gases (WDCGG) (WDCGG, 2017). The trends of mole fractions of $CO_2$, $CH_4$, CO, $H_2$, $N_2O$, and $SF_6$ and the isotopic ratio of $\delta^{13}C$-$CO_2$ and $\delta^{18}O$-$CO_2$ were calculated according to the method of Thoning et al. (1989) with a cut-off frequency of 667 days (0.5472 cycles yr$^{-1}$) for a Fast Fourier Transform (FFT) filter. We also obtained the DMI and ENSO Index from the NOAA/ESRL website (NOAA/ESRL, 2021a; 2021b).





**2.5 Weather data**
Monthly precipitation data for Nainital uses the monthly precipitation of the state of Uttarakhand, which includes
Nainital. The data during January 2007 to December 2017 were taken from the rainfall report on the IMD (India Meteorological
Department) website (http://hydro.imd.gov.in/hydrometweb/(S(fqu5hsvtq3sitn45rjia4qma))/landing.aspx). Monthly
precipitation data for Comilla uses the average monthly precipitation of Eastern Indo-Gangetic Plain in Bangladesh (Rangpur
(25.73°N, 89.23°E and 33 m a.s.l.), Sylhet (24.90°N, 91.88°E and 34 m a.s.l.), Bogra (24.84°N, 89.37°E and 18 m a.s.l.),
Ishurdi (24.13°N, 89.05°E and 13 m a.s.l.), Jessore (23.18°N, 89.17°E and 6 m a.s.l.), Feni (23.03°N, 91.42°E and 6 m a.s.l.),
Barisal (22.75°N, 90.37°E and 3 m a.s.l.), Chattoogram (22.27°N, 91.82°E and 4 m a.s.l.), and Cox's Bazar (21.43°N, 91.93°E
and 2 m a.s.l.)). Data during January 2012 to September 2018 were taken from the JMA (Japan Meteorological Agency)
website (http://www.data.jma.go.jp/gmd/cpd/monitor/climatview/frame.php?y=2019&m=7&d=30&e=0).
**2.6 Back trajectory analysis**
To determine the sources of regional air masses affecting the stations (NTL and CLA), we calculated backward air
trajectories using the Meteorological Data Explorer (METEX) system (Zeng and Fujinuma, 2004) available via the website of
the Center for Global Environmental Research, National Institute for Environmental Studies
(http://db.cger.nies.go.jp/metex/index.html). METEX uses three dimensional wind speed (horizontal and vertical wind )
estimated from the European Centre for Medium Range Weather Forecast (ECMWF) analyses on a $0.5° \times 0.5°$ mesh to
calculate 72-h trajectories. We use 1940m for NTL and 30m for CLA as the starting height.
The ratio of air mass from south was calculated by the frequency of the air mass from south side on the flask sampling
date with reference to the backward air trajectories data.
**2.7 Data analysis method for short-term and long-term**
Mean values for every 10 days were calculated from the weekly data and were used to calculate the long-term trend
and smoothing fitting curve. The value of the missing period was supplemented with an approximate expression of the values
before and after the missing period for calculating the continuous long-term trend and smoothing fitting curve.
Long-term trends of the mole fractions were calculated based on the idea of Thoning et al. (1989) with a cut-off
frequency of 667 days (0.5472 cycles yr$^{-1}$) for a FFT filter. The smoothing fitting curve was made for an FFT filter with a cut-
off frequency of 50 days (7.3 cycles yr$^{-1}$).
We defined and expressed seasonal component by a "$\Delta$" term (e.g., $\Delta CO_2$) which was calculated by subtraction of the
long-term trend curve from 10 days mean of real data. Also, we defined and expressed short-term variations by a "d" term
(e.g., $dCO_2$), which were characterized by the deviation of 10 days mean of real data from the smoothing fitting curve. Figure
2(c) shows how such components were calculated. Growth rates of mole fraction of observed gases were calculated using the
long-term trends.
**3. Results and discussion**
**3.1 Overview of GHGs levels at both sites**
Basically, the air masses over the Indian subcontinent were transported from the Indian Ocean region during summer
(monsoon season) and from the inland during winter. Air mass trajectories are shown for our sampling sites and related sites
in Figure 3. In the case of anthropogenic GHGs, except $CO_2$, their mole fractions at CLA generally showed relatively low
levels when the air mass came from the ocean, while the mole fractions were relatively high when the air mass came from
inland. On the other hand, mole fractions of GHGs at NTL overall did not show relatively low levels, even if the air mass came
from the Indian Ocean region (i.e., south-eastern wind) because the air mass from Indian Ocean was strongly affected by local




GHGs emissions while passing over the Indo-Gangetic Plain. However, the $CO_2$ mole fraction changed not only due to
transport but also due to the photosynthetic sink strength of terrestrial ecosystems and cultivated crops.

Annual mean GHG mole fractions at NTL and CLA are summarized in Table 1. Annual $CO_2$ mole fractions at both sites

were quite low compared to MLO and other Indian sites such as CRI. For example, in 2010, 386.5 ppm was reported at NTL,
391.9 ppm at CRI (Bhattacharya et al., 2009), and 391.3 ppm was reported at PON (Lin et al. 2015). Note that there is no data
for CLA in 2010, however the annual $CO_2$ mole fraction at CLA is usually only 1–2 ppm higher than at NTL. This seemed to
be due to the influence of photosynthesis at both sites. Generally, the $CO_2$ mole fractions at NTL and CLA decreased strongly
(typically twice a year) due to photosynthesis of local crops, making the annual $CO_2$ levels lower than at other sites despite the
likelihood that anthropogenic emission are high in this area.

On the other hand, the annual mean mole fractions of $CH_4$, CO, $H_2$, and $N_2O$ at NTL and CLA (Table 1) were almost at

the highest levels on the Indian subcontinent due to the influence of strong emission sources. For example, the annual mole
fractions of NTL and CLA were 50–470 ppb, 30–200 ppb for CO, and 0–5 ppb for $N_2O$ higher compared to other Indian sites
(e.g., CRI [Bhattacharya et al., 2009], HLE, PON, and PBL [Lin et al., 2015]). In this region, high $CH_4$ and $N_2O$ emissions
were possible from paddy fields and cultivated areas. Also, much CO is considered to be produced by biomass burning in this
region. As for $H_2$, the mole fraction at CLA was higher than those at other Indian sites, however, it was relatively low at NTL
compared to other sites such as CRI (Bhattacharya et al., 2009), PON, and PBL (Lin et al., 2015), but similar to HLE, which
is located on a higher mountain. In the case of the $SF_6$ mole fraction, it has smaller regional differences, suggesting there are
no remarkable $SF_6$ sources near the measurement sites. Below we describe in detail the characteristics of sources and sinks of
each component ($CO_2$, $\delta^{13}C$-$CO_2$, $\delta^{18}O$-$CO_2$ $CH_4$, CO, $H_2$, $N_2O$, and $SF_6$) at NTL and CLA on the Indo-Gangetic Plain in terms
of seasonal variations, amplitudes, and growth rates.

### 3.2 $CO_2$ and $\delta^{13}C$-$CO_2$

#### 3.2.1 $CO_2$ mole fraction and growth rate variations

Figure 4 shows the time series of the atmospheric $CO_2$ mole fraction and the isotopic ratio of $\delta^{13}C$-$CO_2$ at our sampling

sites (NTL and CLA) together with data from CRI on the west coast of India and MLO in Hawaii. The $CO_2$ mole fractions at
NTL and CLA in August–October were characteristically lower (approximate 10–20 ppm) than the mole fractions observed
at CRI and MLO. The CRI and MLO sites are representative of $CO_2$ mole fractions in the Southern and Northern Hemisphere,
respectively, for the period of the southwest monsoon season (June–September). On the other hand, the $\delta^{13}C$-$CO_2$ at NTL and
CLA were inversely correlated with the $CO_2$ mole fractions, and generally the values at both sites were higher than at MLO
and CRI.

Air masses at NTL and CLA in August–October passed over the Indo-Gangetic Plain and the southeast area of India,

respectively, while the air masses of CRI were transported from the Indian Ocean region (Fig. 3). Thus, it was suggested that
the air mass from the Indian Ocean in August–October prevailing over CRI was hardly influenced by anthropogenic emission
and photosynthesis over the Indian subcontinent, whereas $CO_2$ mole fractions over NTL and CLA seemed to be influenced
during these season by the sources and sinks on the Indo-Gangetic Plain and the south/east areas of the Indian subcontinent.
Such transport characteristics must affect the annual average and growth rates in the $CO_2$ molar ratio and $\delta^{13}C$-$CO_2$ in addition
to their seasonal variations.

We show the $CO_2$ growth rates observed at NTL, CLA, and MLO in Figure 5(a). Mean $CO_2$ growth rate at NTL

(approximately 2.0 ppm yr$^{-1}$ during 2007–2020) and CLA (approximately 3.1 ppm yr$^{-1}$ during 2013–2020) were similar to
other sites (e.g., MLO). However, variations of the calculated growth rates were greater than those at MLO. The range was 0–
5 ppm yr$^{-1}$ in the case of NTL, and CLA had higher variability than NTL because local sink and source influences affected the
concentration more than remote sites such as MLO. In general, Pacific sites such as MLO and Japanese remote sites in the
Northern Hemisphere showed a relationship between $CO_2$ growth rates and the ENSO index (e.g., Keeling, 1998). This





relationship is often explained from the viewpoint of a global temperature anomaly, which has a strong relationship with the
ENSO index. On the other hand, the variability at NTL has no associations with the variability in the $CO_2$ growth rate at MLO
and the ENSO index (Fig. 5[b]). Both growth rates seemed to be slightly inversely correlated with each other from 2007 to
2015. However, since then, similar relatively high growth rates have been observed for both sites around 2015-2016 and 2018-
2019, indicating that overall, the $CO_2$ growth rate at NTL is less correlated with the $CO_2$ growth rate at MLO and the ENSO
index.
It is well known that the Indian Ocean Dipole controls meteorological conditions such as air mass transportation and
precipitation patterns on the Indian subcontinent (e.g., Saji et al., 1999, Ashok et al., 2004, Hong et al., 2008). Such changes
in regional climatic pattern could affect the $CO_2$ uptake flux by plants in the surrounding area and the atmospheric movement,
leading to a change in the $CO_2$ growth rate. However, we did not find a simple relationship between DMI and $CO_2$ growth rate
at NTL (Fig. 5[b]). Here we have shown that the pattern of $CO_2$ growth rate in this region is different from the global pattern
seen in places like MLO, but the relationship between local climatic factors and changes in $CO_2$ sinks and emissions is likely
to be complex, and further study is needed to interpret the differences.

### 3.2.2 Seasonal variation and its characteristics

Figure 6(a)-(d) show the seasonal variations in $CO_2$ mole fractions and isotopic ratios of $\delta^{13}C$-$CO_2$ at NTL, CLA, CRI,
and MLO, which were calculated by subtraction of the measured value from the long-term trend. The annual amplitudes of the
$CO_2$ mole fraction (Table 2) at NTL ($22.1 \pm 3.9$ ppm) and CLA ($20.3 \pm 5.7$ ppm) were much larger than those at other Indian
sites (CRI, 15 ppm; HLE, 8.2 ppm; PON, 7.6 ppm; PBL, 11.1 ppm). Also, the annual amplitudes of $\delta^{13}C$-$CO_2$ at NTL ($0.96 \pm$
$0.16$‰) and CLA ($0.85 \pm 0.19$‰) were larger than that at CRI (approximately 0.6‰). These results suggested that the
atmospheric $CO_2$ mole fraction of NTL and CLA were strongly influenced by photosynthesis of local plants in summer and
their respiration in winter, and other anthropogenic emission which were moderated at the other sites by the influence of the
oceanic air.
As shown in Figure 4 (a) and (b) and Figure 6(b) and (d), the seasonal variation pattern at CLA has two lower seasons
in $CO_2$ and two higher seasons in $\delta^{13}C$-$CO_2$ in February–April and July–October. Similarly, in the case of NTL, we sometimes
observed relatively low mole fractions of $CO_2$ in February–March and September, and higher $\delta^{13}C$-$CO_2$. Especially, the $CO_2$
mole fraction at CLA in February–March decreased remarkably, by up to approximately 8 ppm. In general, in many cases
including at MLO, only a summer minimum $CO_2$ mole fraction is observed, while a minimum in February–March is not
usually observed.
Twice-yearly decreases in the $CO_2$ mole fraction have also been observed at several Indian sites such as Dehradun
(northern Indian site; Sharma et al., 2013), Sinhagad (western Ghats site; Tiwari et al., 2014), Ahmedabad (western Indian
site; Chandra et al., 2016), Shadnagar (central Indian site; Sreenivas et al., 2016), and PON (southeast coast Indian site; Lin et
al., 2015), however, these studies did not clearly mention such variations. Umezawa et al. (2016) reported that the decrease in
the $CO_2$ mole fraction near the ground in February–March was caused by photosynthesis of local crops, which was detected
by the vertical $CO_2$ profiles over New Delhi airport. Those sites are located on the Indo-Gangetic Plain or received air masses
passing over the Indo-Gangetic Plain or Indian subcontinent. On the other hand, the decrease in the $CO_2$ mole fraction in
February–March was not detected at CRI (west coast Indian site; Bhattacharya et al., 2009), HLE (northwestern Himalayan
site), or PBL (Andaman Island's site) (Lin et al., 2015). These sites are not located on the Indo-Gangetic Plain. Thus, air
masses at these sites must be mainly transported from the ocean or from areas other than the Indian subcontinent during these
periods.
The characteristic $CO_2$ seasonal variation on the Indo-Gangetic Plain (including NTL and CLA) is very likely to be
related to $CO_2$ uptake by regional vegetation. In the region near NTL, rice, wheat, and other cereals and millets were mainly



cultivated (DAC/MA, 2015; SID/MP, 2018; and DES/MAFW, 2019). Generally, in the case of Uttar Pradesh state located in
the center of the Indo-Gangetic Plain, rice and other summer plants (maize, millets, etc.) are planted mainly in June–July and
harvested in October–November, while large areas of wheat are sown in October–December and harvested in March–April.
Therefore, relatively low $CO_2$ mole fractions observed in those periods are considered to be due to $CO_2$ uptake by plants
cultivated in each season near NTL. Panigrahy et al.(2010)reported the main rice growing seasons in North India to be July–
September and February–March by using the Normalized Difference Vegetation Index (NDVI). Nayak et al. (2010) also
reported that Net Primary Productivity (NPP) on the Indo-Gangetic Plain increased in August–September and February–March,
estimated from the NDVI.

In Bangladesh, rice, being the staple food, is cultivated three times a year in some regions. Usually rice is grown twice

(*Aus* and *Amon* rice) from April–October (including the monsoon season), however, often rice is also cultivated (*Boro* rice) in
the winter season from November–April (SID/MP, 2018). Other agricultural products include maize, jute, and vegetables in
the summer season, and small amount of wheat in the winter season. Therefore, we concluded that the observed lower $CO_2$
mole fractions in July–October and February–March were influenced by $CO_2$ uptake by local plants (mainly rice). Especially
at CLA, the lower mole fraction in February–March was clear and a strong contribution from $CO_2$ uptake from *Boro* rice was
estimated. As another viewpoint on $CO_2$ seasonal variation, we observed that the $CO_2$ maximum in May was not so high, while
the $CO_2$ mole fraction in December was higher. Because precipitation in Bangladesh is stronger than in the north Indian region,
the duration of rice cultivation over summertime is also longer than in north India. Therefore, the contribution of plant uptake
to the $CO_2$ mole fraction in the atmosphere at CLA over the summer season is likely to be relatively large compared to that at
NTL.

Thus, the decreases in the $CO_2$ mole fractions in February–March and September in NTL and CLA were estimated to

be caused by photosynthesis of plants cultivated in each season over the Indo-Gangetic Plain. NTL and CLA indicated this
more clearly compared with other Indian sites due to the proximity to the source region. Figure 7(a) shows the relationships
between the annual mean $CO_2$ mole fraction and $\delta^{13}C$-$CO_2$ in 2010 and 2012. The slope between the $CO_2$ mole fraction and
$\delta^{13}C$-$CO_2$ showed -0.050 and -0.054‰ ppm$^{-1}$ which indicated that the spatial variability of the atmospheric $CO_2$ mole fraction
(e.g., a lower mole fraction at NTL than at MLO and CRI) basically occurred due to $CO_2$ exchange between the atmosphere
and terrestrial biosphere.

Furthermore, we examined the relationship of the $CO_2$ mole fraction and carbon isotope ratio, because there are some

seasonal differences in the species cultivation. On the Indo-Gangetic Plain, rice (especially in Bangladesh) and wheat
(especially in North India), as $C_3$ plants, are cultivated in January–March, while $C_4$ plants (e.g., maize, sugarcane, sorghum
and Bajra (Pearl millet) in addition to rice are cultivated on the Indo-Gangetic Plain and in Bangladesh in June–September
(DAC/MA, 2015; SID/MP, 2018; DES/MAFW, 2019). We calculated the end member of the isotope value for absorbed $CO_2$
by using intercept values of the "Keeling plot" between the reciprocal of the $CO_2$ mole fraction and the ratio of $\delta^{13}C$-$CO_2$
obtained from two continuous datasets of air samples, which has > 1 ppm difference in $CO_2$ mole fraction and > 0.05‰ in
$\delta^{13}C$-$CO_2$. Since in this study two datasets had 1-week intervals, we assumed that the difference in $CO_2$ and $\delta^{13}C$ between two
datasets would include broader influences of photosynthetic activities from relatively large areas on the Indo-Gangetic Plain.

We found that the intercept values of NTL and CLA showed differences in January–March and June–September (Fig.

7[b]), which appeared to reflect the differences in the contributions of $C_3$ and $C_4$ plants in this region. In June–September, we
found relatively heavier intercept values at both NTL (-25.0 ± 2.4‰) and CLA (-23.5 ± 4.1‰), suggesting that $C_4$ plants partly
contributed to the $CO_2$ absorption (or emission) in this season, while in January–March, the end member showed -29.0 ± 4.3‰
(NTL) and -28.3 ± 4.0‰ (CLA), which were similar to the general $C_3$ plant (rice or wheat). If we assume the value for $C_4$ plant
to be -12 to -14‰, the contributions of $C_4$ plant in NTL and CLA were approximately 25 ± 5% and 31 ± 9%, respectively.
According to database (DAC/MA, 2015; SID/MP, 2018; DES/MAFW, 2019) for crops area in Uttar Pradesh district, the area's
ratio of $C_4$ plants (e.g., maize and sugarcane) to $C_3$ plants in the summer season was approximately 26% in 2012, which was





a similar proportion as estimated by the C isotope ratio. In the case of Bangladesh, despite there being no recent data reported,
according to data in 2008, the area for maize was approximately < 10% compared to the rice area. However, based on the
recent C isotope ratio, it appears likely that more maize has been cultivated.
**3.3 $\delta^{18}$O-CO$_2$**

In general, $\delta^{18}$O-CO$_2$ is related to that value of water in plants and soil, because oxygen atom of CO$_2$ can be exchanged

with oxygen atom of H$_2$O in plant and bacteria cells during photosynthesis and soil respiration. Plants and soil water mainly
originate from rainwater in the study region, however, in the case of the agricultural area, water is often introduced by irrigation
systems using river and groundwater. In many cases, photosynthesis produced relatively heavier $\delta^{18}$O-CO$_2$ than soil respiration
because $\delta^{18}$O-H$_2$O in plant becomes heavier than soil water due to plant transpiration.

Larger amplitudes (approximately 3‰) in the seasonal variation of $\delta^{18}$O-CO$_2$ at both NTL and CLA were observed,

compared to that of MLO (approximately 0.4‰) (Fig. 8[a]). The isotopic ratio of $\delta^{18}$O-CO$_2$ at CRI (Bhattacharya et al., 2009)
was reported to have similar seasonal variation (i.e., high in winter [November–February] and low in September) to our sites.
In the Pacific sites like MLO, $\delta^{18}$O-CO$_2$ has a maximum peak from spring to summer when photosynthesis activity become
dominant, while a minimum is seen around fall when the contribution of soil respiration exceeds that of photosynthesis. On
the other hand, Indian subcontinent sites seemed to have fairy different seasonal variation patterns, having a maximum in
January–February, gradually decreasing from March–September/October, and subsequently rapidly increasing (Fig. 8[c] and
[d]). Such seasonal variation may be influenced by photosynthesis and soil respiration in these regions. However, because
many crops are cultivated through the year in these areas (as mentioned in section 3.2), the contribution of photosynthesis to
the seasonal variation may be relatively small. High soil respiration activity in the wet season can contribute a little more than
during the dry season.

On the other hand, seasonal variations in $\delta^{18}$O of rainwater itself seemed to affect $\delta^{18}$O-CO$_2$ through photosynthesis

and respiration processes. For example, Sengupta and Sarkar (2006) showed the $\delta^{18}$O-H$_2$O in rain at New Delhi (western Indo-
Gangetic Plain) had a higher value in March–May and a minimum value in September. Such variation was fairly consistent
with the seasonal variation in $\delta^{18}$O of CO$_2$ at NTL. Similarly, CLA has a minimum $\delta^{18}$O-CO$_2$ in the atmosphere in October,
which was the same month in which the minimum $\delta^{18}$O-H$_2$O was observed in rain in Eastern Indo-Gangetic Plain areas (e.g.,
Kolkata [near Bangladesh; Sengupta and Sarkar, 2006] and Cherrapunij [Eastern Indo-Gangetic Plain; Breitenbach et al.,
2010]). During the rainy season, due to the so-called "amount effect", $\delta^{18}$O-H$_2$O in rain will decrease with an increase in the
amount of precipitation (e.g., Rozanski et al., 1993). However, in the Indian region it has been reported that seasonal changes
in the origin of moisture strongly affected the $\delta^{18}$O-H$_2$O (Sengupta and Sarkar, 2006, Tanoue et al., 2018). In winter (i.e., when
there is less rain), moisture comes from the west or north. Therefore, the northern area of the Arabian Sea and the western land
area supply moisture, which has a higher $\delta^{18}$O-H$_2$O. However, the air mass in the summer monsoon season (mainly June–
September) comes from the southern part of the Arabian Sea and sometimes passes over the Bay of Bengal carrying much
moisture. The value of $\delta^{18}$O-H$_2$O in the moisture in the air mass decreases with the process of raining along the air trajectory.
In the post-monsoon season (mainly October–December), some portion of moisture comes from the Pacific, Bay of Bengal,
and the inland area (Tanoue et al., 2018).

In the winter monsoon season (mainly February–May), $\delta^{18}$O-H$_2$O in rain was reported to be approximately 0–1‰ (vs

VSMOW). During the winter monsoon season, there is little precipitation, so plant cultivation utilizes irrigation systems using
river and groundwater. River and groundwater usually show not so large seasonal variation in $\delta^{18}$O and have a close value to
the annual mean of $\delta^{18}$O-H$_2$O in rain, such as -6 to -8‰ (Kumar et al., 2019). According to the variation of $\delta^{18}$O-CO$_2$, in winter
its value was approximately 2‰ (vs VPDB-CO$_2$; VPDB-CO$_2$ scale is fairly close to the scale of CO$_2$ equilibrated with
VSMOW water as mentioned in section 2.3), which was higher than that of rain and other water reservoirs, suggesting that
$\delta^{18}$O-H$_2$O in plants and soil must become higher due to transpiration during dry and relatively warm conditions in winter.





Based on the fact that during the summer monsoon season, $\delta^{18}O\text{-}CO_2$ decreased from 1 to -2‰ with a decrease of $\delta^{18}O\text{-}$
$H_2O$ from 0 to -10 or -15‰ in the rain, the range of variation in $\delta^{18}O\text{-}CO_2$ was approximately one third or one fifth that of rain.
Because land water may come from both rain and irrigation systems, the real ranges of $\delta^{18}O$ in soil water and plant water are
likely to be smaller than in the case of rain only. Furthermore, because $CO_2$ from soil respiration contributes more in the rainy
season, a balance between photosynthesis and respiration $CO_2$ will, in general, have a small effect on the seasonal variation.
As for the annual trend of $\delta^{18}O\text{-}CO_2$ shown in Figure 8(b), NTL showed a similar pattern to that of MLO whereas CLA
showed a different trend. The $\delta^{18}O\text{-}CO_2$ at NTL began at 0.8‰ in 2007, decreased to 0.2‰ in 2011, then again became heavier
(toward 1.0‰) during 2014–2016 (Fig. 8[b]). In northern India, relatively high precipitation was reported during 2011–2013.
The tendency of lower $^{18}O\text{-}CO_2$ may have some relationship with the amount of precipitation. In 2008 and 2016 considerable
amounts of precipitation fell near NTL. The $^{18}O\text{-}CO_2$ level also seemed to become relatively low. A La Nina event occurred
from late 2010 to 2012 and the amount of precipitation increased worldwide from 2010 to 2013. Such large-scale climatic
effects are very likely to affect the $^{18}O\text{-}CO_2$ level observed at MLO. In the case of CLA, precipitation increased in 2015–2017
(rather than in 2011–2013) and the $^{18}O\text{-}CO_2$ level at CLA seemed to become lower at that time with the increase of precipitation.
Analyzing the relationship between the monthly amount of precipitation and $\delta^{18}O\text{-}CO_2$ in Figure 8(e) and (f), a weak negative
correlation can be seen. Therefore, the amount of precipitation partly contributes to the regional level of $\delta^{18}O\text{-}CO_2$. However,
it must be influenced not only by precipitation but also by seasonal changes in air flow patterns and rain systems, as explained
above, as well as by the water reservoir situation, soil water content at that time, and photosynthesis in the region.
If the ground water storage decreases due to wider usage of irrigation and/or less precipitation in recent times, it causes
a stronger transpiration effect in the soil environment, making the $\delta^{18}O$ of soil water heavier than usual. Roxy et al. (2015) and
Asoka et al. (2017) reported that precipitation over the Indian subcontinent and groundwater storage in northern India has had
a decreasing trend due to Indian Ocean warming, which is estimated to have occurred due to the weakening trend of the
summer monsoon cross-equatorial flow (Swapna et al., 2014). However, much longer records of $CO_2$ isotopic ratios are needed
to clarify the increasing trend in $\delta^{18}O\text{-}CO_2$ and the relationship with climatic changes in this region.
**3.4 CH$_4$**
The $CH_4$ mole fractions at NTL and CLA are illustrated in Figure 9(a). We detected high $CH_4$ mole fractions at NTL
and CLA, where they sometimes exceeded 2,100 and 4,000 ppb, respectively, showing that the Indo-Gangetic Plain region
had relatively strong $CH_4$ emissions. The seasonal amplitude of the $CH_4$ mole fraction, especially at CLA (486 ± 225 ppb;
Table 2) was much larger than the those of other Indian sites such as NTL (114 ppb), CRI (200 ppb) (Bhattacharya et al., 2009),
Darjeeling (400 ppb) (Ganesan et al., 2013), HLE (29 ppb), PON (124 ppb), and PBL (144 ppb) (Lin et al., 2015), which
indicated that the contribution of the $CH_4$ source (e.g., rice cultivation) around CLA was relative strong.
Mean seasonal variations in the $CH_4$ mole fraction for both sites were calculated and are shown in Figure 9(c) and (d).
The mole fractions at both NTL and CLA had the highest peak in August–October and a small peak in March. In general, the
$CH_4$ mole fraction in the Northern Hemisphere decreased in July–September (summer season) through the decomposition
process by reaction with OH radicals during this period. A higher $CH_4$ mole fraction in this period strongly suggests that there
are some sources of $CH_4$. Observation results at Darjeeling (north-eastern Indian site; Ganesan et al., 2013), HLE (Lin et al.,
2015), and Shadnagar (Sreenivas et al., 2016) also indicated high $CH_4$ mole fractions during August–October. Ganesan et al.
(2013) reported that the $CH_4$ mole fraction at Darjeeling was enhanced by transported air masses from the Indo-Gangetic Plain.
Lin et al. (2015) and Sreenivas et al. (2016) showed that the high $CH_4$ mole fractions at HLE and Shadnagar were influenced
by emissions from paddy fields and wetlands. Garg et al. (2011) showed that $CH_4$ emission from rice fields was estimated to
be approximately 17% of the total $CH_4$ emissions in India. According to the emission database of EDGAR v4.3.2 (EC-
JRC/PBL, 2016), rice cultivation was the largest source of $CH_4$ (approximately 50%) in Bangladesh.





Bhatia et al. (2011) measured the $CH_4$ flux from paddy fields at New Delhi and showed that it was the highest in August–September due to the increase in the activity of rice roots and bacteria in the paddy field soils. Ali et al. (2012) also measured the $CH_4$ flux from paddy fields at Bangladesh and reported that the $CH_4$ flux was maximized within 77–98 days after the planting of rice due to the increase in root respiration and carbon in soil. It was considered that both March and September–October were consistent with the timing of increasing $CH_4$ production at rice fields according to the customary cultivation schedule of rice in this region.  In Bangladesh and the eastern Indian district, rice is cultivated from November– September, as mentioned above in the $CO_2$ section, and $CH_4$ emissions are considered to continue during winter, supporting higher $CH_4$ mole fractions from August–March, especially at CLA.

On the other hand, CRI (Bhattacharya et al., 2009), PON, and PBL (Lin et al., 2015) did not show higher $CH_4$ mole fractions in August–October, as shown in Figure 9(c) and (d). The air masses at those sites in August–October were transported from the Indian Ocean, which may have only a minimal influence from agricultural emission.

$CH_4$ mole fractions at NTL and CLA were higher than that at MLO, even at the time of year when rice is not cultivated. $CH_4$ emissions from the enteric fermentation and wastewater handling were reported to be large sources according to the emission database in EDGAR v4.3.2 (EC-JRC/PBL, 2016). Garg et al. (2011) reported that enteric fermentation by cattle and buffalo contributes approximately 40% emissions in India. Such $CH_4$ emissions must always elevate the $CH_4$ mole fraction in the air mass in these sites regardless of the season.

In addition, biomass burning (including residential cooking and agricultural residue burning) is very likely to have contribution to the $CH_4$ mole fraction according to the inventory evaluation (i.e., 21% contribution; Garg et al, 2011). Reasonably good correlations were seen between short term components in variations of $CH_4$ and CO in January–March, April–June, and October–December. Ratios of $dCH_4$ to $dCO$ showed ranges such as 0.64–0.80 ppb ppb$^{-1}$ in NTL and 1.85– 1.98 ppb ppb$^{-1}$ in CLA, as shown in Figure. 9(e) and (f). One of the major CO sources in India was considered to be biomass burning (Dickerson et al., 2002). Akagi et al (2011), EC-JRC/PBL (2016), and Sfez et al. (2017) reported that the emission ratios of $CH_4$ to CO in biomass burning such as crop residue burning, firewood burning, and biogas burning were 0.04–0.90 ppb ppb$^{-1}$. Therefore, the ratios observed in these seasons could suggest a strong influence on $CH_4$ and CO emissions from biomass burning (such as crop residue burning), despite the other large $CH_4$ emissions such as paddy fields and waste treatment, which will increase the ratio, especially at CLA in July–September.

As a result, it is evident that annual $CH_4$ mole fractions at the sites used in this study on the Indo-Gangetic Plain are enriched by various $CH_4$ sources, depending on the season. Generally speaking, because April–June is a dry and hot season, $CH_4$ decomposition processes will proceed, decreasing its mole fraction at both sites.

The variability in the $CH_4$ growth rate in the trend line at NTL was different to the variability at MLO (Fig. 9[b]), which may be influenced by regional climatic condition, including the Indian Ocean Dipole. Because the frequency of air mass transportation from the south increased if the Indian Ocean Dipole was often activated, the air mass passed over the Indo-Gangetic Plain (which has strong $CH_4$ emissions), reaching NTL with a high $CH_4$ mole fraction. The difference between the variability in the $CH_4$ growth rate between NTL and CLA may also be explained by the above hypothesis. If the frequency of air mass transportation from the south increased by the activation of Indian Ocean Dipole (e.g., in 2015) because the air mass was directly transported from the Indian Ocean with a relatively low $CH_4$ mole fraction, the $CH_4$ mole fraction at CLA would become relatively low compared to a usual year (Fig. 9[b]). On the other hand, as mentioned previously, in 2015–2017, even in high Indian Ocean Dipole mode, Bangladesh had relatively high precipitation which could strengthen $CH_4$ production from rice paddy fields and other aquatic environments. This potential situation well-matched the high $CH_4$ mole fraction in summer and the high growth rate at CLA during 2016–2017.





**3.5 CO**
High annual CO mole fractions at both NTL and CLA (Table 1) indicated that the atmosphere over the Indo-Gangetic
Plain was influenced by strong CO emission sources such as burning of harvest residues and residential burning using solid
biofuel, which are considered to be main CO emission sources in the region (EC-JRC/PBL, 2016). However, of course, CO
originating from car exhaust and industrial activities remains very likely to have made some contributions to the CO mole
fraction (EC-JRC/PBL, 2016).
The main crops around NTL are rice and wheat and the harvesting periods are September–November and April–May,
respectively (DAC/MA, 2015). Farmers in this area generally burn harvest residues at their farmland after harvest (Lohan et
al., 2018). Venkataraman et al. (2006) reported that the amount of burning on the Western Indo-Gangetic Plain has two peaks
annually, i.e., in May and November. We could observe the same seasonal variation (i.e., two mole fraction peaks in May and
November) in the CO mole fraction in the atmosphere at NTL (Fig. 10[c]). Sharma et al. (2010) suggested that the high CO
mole fraction on the Western Indo-Gangetic Plain is emitted in October by the burning of harvest residues, based on data from
satellite observations. Kumar et al. (2011) also reported that the highest densities in fire spots were seen in spring and autumn
on the western Indo-Gangetic Plain. These suggested that CO emissions from the burning of harvest residues was one of the
most important sources on the Western Indo-Gangetic Plain in these seasons.
On the other hand, the seasonal variation in CO mole fraction at CLA exhibited only one peak in October–March (Fig.
10[d]). Such seasonal variation was also detected at CRI (Bhattacharya et al., 2009), PON, PBL (Lin et al., 2015), and
Ahmedabad (Chandra et al., 2016). In Bangladesh, after the end of the monsoon (October–March), harvest residues are burnt
and used to make bricks using some kinds of biofuel as a heat source (Guttikunda et al., 2012). Also, dung is burnt for the
stove (Venkataraman et al., 2010) during the winter season. In addition, biofuel is used for cooking (Lawrence and Lelieveld,
2010) throughout the year. Those activities could emit large amounts of CO (Streets et al., 2003; Venkataraman et al., 2010;
Maithel et al., 2012).
In addition, the seasonal amplitude of the CO mole fraction (Table 2) at CLA ($356 \pm 90$ ppb) on the Eastern Indo-
Gangetic Plain site was much larger than that observed in other Indian sites (e.g., CRI [200 ppb], PON [78 ppb], PBL [144
ppb], and Ahmedabad [270 ppb]). The highest CO amplitude observed at CLA was consistent with the model estimation of
CO emissions, which showed that the Eastern Indo-Gangetic Plain included areas with the highest CO emissions (Kumar et
al., 2013).
On the other hand, the annual mean CO mole fraction at NTL gradually decreased approximately by 50 ppb for 10
years (2006–2015; Fig. 10[a]). Especially, the monthly mean CO mole fraction in November of each year (i.e., the highest
level in the year) at NTL decreased by 120 ppb during that period. This suggests that the amount of harvest residues burnt
decreased, the ratio of incomplete combustion in car engines was improved, or the type of fossil fuel for cooking changed from
biofuel to natural gas. Such decreasing trends in the CO mole fraction level were also detected by Pandey et al. (2017) who
reported total-column CO levels during 2003–2014 over the Indo-Gangetic Plain. However, the CO mole fraction level at NTL
appeared to increase slightly from 2015. Although the reason for the increase is unclear from this study only, CO emissions
from car exhaust were recently estimated to have increased (EC-JRC/PBL, 2016). Therefore, further monitoring is important.
The trend in the CO mole fraction and its inter-annual variability at NTL was similar to those in $CH_4$ at NTL (Fig. 9[b]
and Fig. 10[b]). The mole fractions of CO and $CH_4$ at NTL tended to be slightly higher when the air mass passed over the
Indo-Gangetic Plain, where there are strong sources of both CO and $CH_4$. In 2015 and 2017, a large positive Indian Dipole
Mode occurred, in addition to El Nino in 2015. Therefore, we observed more frequent southern winds, causing higher $CH_4$
and CO mole fractions at NTL. However, at CLA, southern wind will decrease the mole fraction of CO. Thus, temporal
variations of both CO and $CH_4$ mole fractions in both sites must be strongly controlled by meteorological conditions as well
as source strength.





**3.6 H$_2$**

491  Mole fractions, growth rates, and seasonal variations of H$_2$ at both sites are shown in Figure 11(a-d). It was found

492 that CLA, especially, showed a higher mole fraction than the other sites. Novelli et al. (1999) reported that the mainly sources

493 of H$_2$ were combustion (fossil fuel combustion and biomass burning) and photochemical sources such as the oxidation of CH$_4$

494 and non-CH$_4$ hydrocarbons (NMHCs), which account for 90% of the total source. The other 10% is attributed to emissions

495 from volcanoes, oceans, and nitrogen fixation by legumes. Therefore, we have to assume that there are some emission sources

496 at CLA.

497  On the other hand, H$_2$ is removed from the troposphere by reacting with OH and by deposition and oxidation at

498 surface soil. The amounts of sources and sinks for H$_2$ in the global budget were estimated to be equal, resulting in a near-

499 equilibrium state (Novelli et al., 1999). The strengths of H$_2$ removal in the atmosphere over the Indian subcontinent do not

500 differ greatly by region according to Yashiro et al. (2011), whereas the strengths of H$_2$ sources may differ by region (Price et

501 al., 2007). Lin et al. (2015) reported that H$_2$ mole fractions at Indian sites were influenced by biomass burning and were 0–40

502 ppb higher than those at regional background sites (e.g., eastern Kazakhstan and central China). Figure 11(c) and (d) show the

503 seasonal variations of the H$_2$ mole fraction at NTL and CLA, which illustrate the maximum in May and the minimum in

504 December at NTL, and the maximum in November–January and the minimum in June–August at CLA, which were different

505 from the averaged seasonal variation in the Northern Hemisphere, which showed the maximum in March–April and the

506 minimum in August–September (Novelli et al., 1999).

507  Because the burning of biomass (such as harvest residuals and dung) appeared to be actively carried out on the Indo-

508 Gangetic Plain (including at NTL) during April–May and at CLA during November–February, H$_2$ production must, therefore,

509 increase during these seasons. Furthermore, since higher CH$_4$ mole fractions at NTL and CLA were observed during August–

510 September and September–October due to strong paddy field emissions at those times, H$_2$ production from CH$_4$ degradation

511 can also increase. Figure 11(e) and (f) show short-term variable components (such as dCO and dH$_2$, and dCH$_4$, and dH$_2$) at

512 both NTL and CLA during those periods, and that they had positive correlations. These figures may suggest some relationship

513 between H$_2$ emission with biomass burning, and between photochemical reactions between OH and CH$_4$, respectively.

514 Furthermore, the minimum H$_2$ in June–August was influenced by a fresh air mass from the Indian Ocean which is only

515 minimally affected by anthropogenic emission.

516  As mentioned above, the H$_2$ mole fraction level at CLA was higher than that at NTL. The amplitude of the seasonal

517 variation of the H$_2$ mole fraction (Table 2) at CLA showed 70.4 ± 42.2 ppb, which was also larger than the amplitudes at other

518 Indian sites such as Nainital (50 ppb), CRI (50 ppb) (Bhattacharya et al., 2009), HLE (22 ppb), PON (16 ppb), and PBL (22

519 ppb) (Lin et al., 2015). These tendencies were consistent with the results of Price et al. (2007), which indicated a larger H$_2$

520 emission area around the Eastern Indo-Gangetic Plain, such as at CLA, than on the Western Indian subcontinent. Thus, our

521 observation and previous studies both indicated that the Indian subcontinent had relatively strong H$_2$ sources.

**3.7 N$_2$O**

523  Garg et al. (2012) reported that the agricultural sector accounted for approximately 75% of the total N$_2$O emission in India

524 in 2005, including around 49% from nitrogen fertilizer use. In particular, they reported that northern India (the Indo-Gangetic

525 Plain) has the highest N$_2$O emission in India because nitrogen fertilizer was applied to extensive paddy fields, was denitrified,

526 and N$_2$O was produced and emitted to the atmosphere. Ganesan et al. (2013) reported that the N$_2$O mole fraction at Darjeeling

527 (north-eastern Indian site) was enhanced due to air mass transportation from the Indo-Gangetic Plain. The annual mean N$_2$O

528 mole fraction at NTL (Table 1) appeared to be almost the same as at Darjeeling sites in North India and was higher than at

529 another two Indian sites (CRI [Bhattacharya et al., 2009] and HLE [Lin et al., 2015]) and at MLO (Fig. 12[a]).

530  Thompson et al. (2014) estimated that the N$_2$O emissions of the Eastern Indo-Gangetic Plain, including CLA, were

531 higher than those of the Western Indo-Gangetic Plain. This is supported by our observation results that show that the N$_2$O





annual mean mole fraction during 2013–2019 at CLA on the Eastern Indo-Gangetic Plain was 1–2 ppb higher than at NTL on
the Western Indo-Gangetic Plain (Table 1), and the seasonal amplitude of the $N_2O$ mole fraction (Table 2) at CLA (4.25 ± 1.45
ppb) was higher than the amplitudes at other Indian sites (NTL, CRI [Bhattacharya et al., 2009], HLE, PON, and PBL [Lin et
al., 2015]). Raut et al. (2011) reported the highest $N_2O$ emission rates in the regions of Bangladesh and Sri Lanka due to their
high usage of urea as a fertilizer.
However, interestingly, PON and PBL, where oceanic air from the Bay of Bengal affected the sites (Lin et al, 2015)
seemed to have relatively higher mole fractions than the sites in this study. As for the seasonal variation in the $N_2O$ mole
fraction at NTL, a higher mole fraction was seen in May–September (Fig. 12[c]). Generally, nitrogen fertilizer was frequently
applied to paddy fields in May–September in northern India. Gupta et al. (2016) measured the $N_2O$ flux in paddy fields at New
Delhi and reported that the flux increased immediately after the application of nitrogen fertilizer to the fields. Therefore, high
$N_2O$ levels and increases in the $N_2O$ mole fraction at NTL in May–September were influenced by the enhancement of the $N_2O$
flux due to the denitrification of nitrogen fertilizer in paddy fields.
The $N_2O$ mole fraction at CLA increased in November–February (Fig. 12[d]) and such seasonal variation was almost
identical to the seasonal variation in CO at CLA. The seasonal component in the $N_2O$ mole fraction ($\Delta N_2O$ = deviation of $N_2O$
mole fraction from the long-term trend) at CLA showed positive correlations ($R^2$ = 0.81–0.88) with that of the CO mole fraction
($\Delta CO$) each year (Fig. 11[e]). Also, their ratio ($\Delta N_2O/\Delta CO$) showed 0.013–0.015 ppb ppb$^{-1}$, which was same (0.015 ppb ppb$^{-1}$
) as the ratio of total $N_2O$ and total CO emissions in Bangladesh from the EDGAR v4.3.2 database (EC-JRC/PBL, 2016).
Although such seasonal variation is likely to be partly related to the lower mixing height in the winter season, variations in
$N_2O$ emission flux must affect the seasonal variations in the mole fraction. In general, the CO mole fraction was influenced by
biomass burning in this season. Because many inventory data showed that biomass burning produced both $N_2O$ and CO, $N_2O$
may be affected partly emitted from biomass burning. However, the emission ratios of $N_2O$ to CO are fairly variable with an
approximate range of 0.0004–0.017 (Andreae and Merlet, 2001; Sahai et al., 2007, 2011; EDGAR v4.3.2 [EC-JRC/PBL,
2016]). It seemed that this ratio changes with the type of plants that are burnt. According to Sahai et al. (2011), because the
ratio was approximately 0.004 in the case of rice straw, some portion (e.g., 0.004/0.015, i.e., approximately 27% at the most)
of $N_2O$ in the atmosphere may originate from biomass burning. In addition, since Venkataraman et al. (2010) reported that
dung burning is one of major $N_2O$ sources among many kinds of biomass burning in India, its contribution was also possible.
On the other hand, nitrification and denitrification processes of nitrogen fertilizer in rice paddy soil are considered
to be major causes of $N_2O$ emissions in this region (EDGAR v4.3.2), however, the emission rate appeared to have seasonal
variation. Related to the irrigation system, the $N_2O$ flux was thought to be larger in alternating wet and dry conditions than
under continuously flooded conditions (Akiyama et al., 2005; Gaihre et al., 2018; Begum et al., 2019). In the summer monsoon
season, many rice paddies fields in Bangladesh must have enough water level because of the ample amount of precipitation.
After the summer monsoon (from October), the water level in the paddy field intermittently changed with the situation.
Therefore, relatively a higher $N_2O$ emission rate likely occurred during the winter season, when rice (*Boro* rice) was still grown,
enhancing the $N_2O$ mole fraction in the winter season. Further observations of high frequency variations of both $N_2O$ and CO
mole fractions will contribute towards precisely evaluating the $N_2O$ emission sources at this site.
The $N_2O$ growth rates at NTL and CLA were similar to that of MLO (Fig. 12[b]), however, the variations in the $N_2O$
growth rate at both NTL and CLA were larger than that of MLO during 2016–2020. The variation in the $N_2O$ growth rate
showed a similar pattern to the growth rates of CO and $H_2$ (Fig. 9[b] and Fig. 10[b]), indicating that the sources of these gases
had basically common characteristics.
**3.8 SF₆**
$SF_6$ is mainly emitted artificially from factories and urban areas (Olivier et al., 2005). Ganesan et al. (2013) reported
that the $SF_6$ emission at Darjeeling (northeastern Indian site) was considerably weak. Our results also showed that $SF_6$ mole





fractions at NTL and CLA were almost the same as the background $SF_6$ mole fraction (e.g., MLO in Fig. 13[a] and other sites
such as HLE, PON, and PBL [Lin et al., 2015]). In addition, the annual amplitudes of the $SF_6$ mole fraction at Indian sites
(HLE, PON, and PBL) were 0.15, 0.24, and 0.48 ppt, respectively, which were almost within the same range (0.15–0.23 ppt)
as at NTL and CLA (Table 2). These results suggested that there was no large $SF_6$ source on the Indo-Gangetic Plain.

Figure 13(c) and (d) show that the seasonal variations of the $SF_6$ mole fraction at NTL and CLA decreased in summer

(NTL: July, CLA: June–August), which was the same variation as those detected at PON and PBL (Lin et al., 2015). In the
summer season, air masses from the south via the Indian Ocean prevailed in the NTL and CLA regions, as shown in Figure 2.
Generally, the $SF_6$ mole fraction in the Southern Hemisphere was lower than that in the Northern Hemisphere (Geller et al.,
1997). Thus, the seasonal variation in the $SF_6$ mole fraction was explained by the frequency of air mass transportation from
the south.

Figure 13(b) shows the interannual variability of the $SF_6$ growth rate at NTL, CLA, and MLO and southern air mass

contribution at NTL and CLA. The variability in the $SF_6$ growth rate at NTL was different to the variability at MLO, and in
fact we could see an anticorrelation between them. In the case of CLA, an anticorrelation was not so clear because of a relatively
shorter data record. The decrease in the growth rate at NTL seemed to have a relationship with the increase in the frequency
of southern air mass transportation. This indicated that the growth rate of the $SF_6$ mole fraction at NTL may be controlled by
the regional climatic condition though the transportation process. Because $SF_6$ had weaker sources in Northern India, the
variation in its trend could be explained more clearly by the influence of the air mass movements.

As mentioned above, anticorrelation in the growth rates between MLO and this region was also seen in $CO_2$ and $CH_4$.

Therefore, we must take into consideration the influence of the variation in large-scale atmospheric circulation to the GHG
mole fraction and trends in their growth rates in the Indian region.
**4. Conclusions**

We characterized GHGs and related gases over the Northern Indian region using air samples collected weekly at

Nainital, India (NTL), and Comilla, Bangladesh (CLA), since 2006 and 2012, respectively. Observation data at both NTL and
CLA were compared with the GHG data of other Indian sites and Mauna Loa, Hawaii (MLO) in the Pacific station. From this
comprehensive analysis, it was found that the feature of seasonal and long-term variations in each gas were influenced by the
local sinks and sources during each season, and annual climatic conditions on the Indo-Gangetic Plain. They were considerably
different to those of the MLO in the Pacific region.

On the Indo-Gangetic Plain, rice, wheat, other cereals, and millet are cultivated in the respective seasons corresponding

to the change between wet and dry climatic conditions. Therefore, seasonal variations in the atmospheric $CO_2$ mole fraction
were strongly influenced by the crop $CO_2$ sink at that time. In general, low $CO_2$ mole fractions in the winter season in the
Northern Hemisphere were not observed, however, we observed relatively lower mole fractions during January–March in this
region, especially at CLA. In Bangladesh, rice is grown even in the winter season. The $\delta^{13}C\text{-}CO_2$ signature showed $C_3$ plants
(e.g., rice and wheat) affected the $CO_2$ mole fractions in the winter season, while in the summer season the $\delta^{13}C\text{-}CO_2$ signature
showed $C_4$ plants (corn, sugar cane etc.) contributed some portion.

The seasonal variations in $\delta^{18}O\text{-}CO_2$ showed almost the same variation as that in the $\delta^{18}O$ in local rain. Effects of the

amount of precipitation and the origin of moisture, appeared to affect $\delta^{18}O$ in local rain and $CO_2$. As a result, $\delta^{18}O$ in $CO_2$ was
affected by the climatic variation related to the amount of precipitation, which was enhanced during 2015–2017. These facts
are also consistent with the explanation that $CO_2$ exchange by photosynthesis (and respiration) by land biomass strongly
affected $CO_2$ seasonality in mole fraction.

At both sites, higher $CH_4$ mole fractions were observed than were recorded at other Indian sites. Especially, higher

mole fractions than 4000 ppb were recorded at CLA, where rice paddy fields covered the area. Rice cultivation was one of





major emission sources in this region. Because $CH_4$ production activities increased after rice planting, we observed the highest
peak in September–October at both sites and a small peak in spring at CLA. A large amount of precipitation during those
seasons is likely to have affected the $CH_4$ production rate of rice paddy fields through soil anaerobic conditions and, as a result,
increased the atmospheric $CH_4$ mole fraction. Air mass transport also influenced seasonal variation and the variability of its
growth rate. Beside emissions from rice paddy fields, we identified the relationship between biomass burning and the $CH_4$
mole fraction in a season other than September–October, when biomass burning occurred frequently. In addition, enteric
fermentation and wastewater handling were large emission sources in this region. The large number of sources appeared to
increase the average $CH_4$ mole fraction in this region.
CO was strongly related to biomass burning activities at both sites. The mole fraction was high in the dry season and
after crop harvesting. At CLA in winter, a higher mole fraction was observed together with a high $N_2O$ mole fraction, which
may suggest some link to biomass burning as a $N_2O$ source. The CO level gradually decreased throughout the observed period.
CO emissions must, therefore, be reduced by various technical progresses including automobile emission and industrial
combustion efficiency improvements.
We observed higher $N_2O$ levels in the crop season (i.e., the rainy season) from May–September at NTL, but much
higher levels in the winter season at CLA. $N_2O$ is known to be mainly emitted from soil though nitrogen fertilizer applications
to rice fields and crop lands in this region. However, for CLA, we estimated seasonal variations in the emission rate due to the
water level in the rice paddy field, because intermittent irrigation in winter generally produces more $N_2O$ than continuously
flooded conditions in the rainy season.
$H_2$ showed some relationship to both CO and $CH_4$ mole fractions. We found that CO had a good correlation with $H_2$ in
the biomass burning season, indicating some $H_2$ contribution from biomass burning. On the other hand, in the season when the
$CH_4$ mole fraction was high, the $H_2$ mole fraction was also relatively high compared to $CH_4$, suggesting that chemical reactions
of $CH_4$ and $H_2$ may contribute some portion of the $H_2$ mole fraction.
$SF_6$ showed consistent mole fractions with other Indian sites. Seasonal variations were strongly related to the southern
air mass frequency, because the $SF_6$ mole fraction in the southern region was relatively low.
We found that the interannual variabilities in $CH_4$, $SF_6$ and also partly in $CO_2$, growth rates at NTL were anticorrelated
with those at MLO, which is located in the Pacific. Growth rates for many GHGs are known to be influenced by El Nino events
for many reasons (e.g., hot climate, dry conditions on a global scale). However, in the Indian region, growth rates of some
GHGs seemed to be more affected by the regional climate condition, which usually affects air circulation and precipitation in
the Indian region. In the case of CLA, although the data duration was insufficiently short, growth rates of $CO_2$, $CH_4$, and $SF_6$
changed differently from those at MLO, which could be partly explained by the climatic variations. Because CLA is located
relatively close to the ocean, sometimes the variation was thought to be different from that at NTL.
These findings have not been reported previously. In this study, long-term records of GHGs data at NTL enabled a
long-term analysis. These findings suggested that the mole fractions of GHGs and their emissions on the Indian subcontinent
could change with climatic conditions in this region in the near future, in addition to changes in anthropogenic activities
relating to GHG emissions and countermeasure for the emissions. Therefore, long-term GHG monitoring should be continued
and the effectiveness of countermeasures for reducing GHG emissions on the Indian subcontinent, including the Indo-Gangetic
Plain, should be evaluated.
**5. Data availability**
We will add digital object identifiers (DOIs) to weekly flask sampling data of Nainital and Comilla and those data on
our website (http://db.cger.nies.go.jp/portal/geds/atmosphericAndOceanicMonitoring) by 2021.



## Conflicts of Interest

The authors declare no conflicts of interest.

## Acknowledgments

We would like to thank Deepak Singh Chausali and other staff of the Aryabhatta Research Institute of Observational Sciences (ARIES), and the staff of Comilla weather station in Bangladesh Meteorological Department (BMD) for their great support in this project. The establishment and running of the air sampling program were partly supported by the Asia Pacific Network (grant ARCP2011-11NMY-Patra/Canadell), and the Environment Research and Technology Development Fund (JPMEERF20152002 and JPMEERF20182002) of the Ministry of the Environment, Japan and the Environmental Restoration and Conservation Agency of Japan. We thank Pieter Tans, Ed. Dlugokencky, Paul. C. Novelli Geoff. Dutton, Bradley Hall and the Earth System Research Laboratory team of the National Oceanic and Atmospheric Administration (NOAA), and James White, Bruce Vaughn and Sylvia Michel and the Institute of Arctic and Alpine Research team of the University of Colorado for providing the data of Mauna Loa Observatory.

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





**Tables**

Table 1. Annual mean atmospheric mole fractions of $CO_2$, $CH_4$, CO, $H_2$, $N_2O$, and $SF_6$ and isotopic ratio of $\delta^{13}C\text{-}CO_2$ and
$\delta^{18}O\text{-}CO_2$ at Nainital (NTL) and Comilla (CLA) in 2007–2020.

| Site | Year | $CO_2$ ppm | | $CH_4$ ppb | | CO ppb | | $H_2$ ppb | | $N_2O$ ppb | | $SF_6$ ppt | | $\delta^{13}C\text{-}CO_2$ ‰ | | $\delta^{18}O\text{-}CO_2$ ‰ | |
|---|---|---|---|---|---|---|---|---|---|---|---|---|---|---|---|---|---|
| | | Ave | S.D | Ave | S.D | Ave | S.D | Ave | S.D | Ave | S.D | Ave | S.D | Ave | S.D | Ave | S.D |
| Nainital | 2007 | 380.6 | 9.6 | 1928.4 | 70.6 | 238.7 | 100.5 | 546.1 | 19.7 | 321.9 | 0.83 | 6.25 | 0.17 | -8.14 | 0.44 | 0.72 | 1.09 |
| Nainital | 2008 | 383.2 | 7.8 | 1931.0 | 75.5 | 225.4 | 99.4 | 551.8 | 24.1 | 323.0 | 0.83 | 6.57 | 0.29 | -8.15 | 0.35 | 0.50 | 1.00 |
| Nainital | 2009 | 383.5 | 9.3 | 1919.4 | 63.3 | 210.2 | 79.2 | 538.8 | 28.0 | 323.7 | 0.88 | 6.95 | 0.28 | -8.13 | 0.44 | 0.55 | 0.87 |
| Nainital | 2010 | 386.5 | 9.0 | 1925.7 | 59.7 | 214.4 | 92.6 | 537.9 | 25.6 | 324.7 | 0.87 | 7.19 | 0.24 | -8.19 | 0.42 | 0.28 | 1.13 |
| Nainital | 2011 | 389.6 | 6.3 | 1945.2 | 70.3 | 213.7 | 72.1 | 544.6 | 24.5 | 325.4 | 0.97 | 7.52 | 0.21 | -8.28 | 0.32 | 0.35 | 1.20 |
| Nainital | 2012 | 391.2 | 7.5 | 1956.0 | 76.7 | 222.1 | 79.3 | 552.6 | 29.9 | 326.2 | 1.18 | 7.85 | 0.35 | -8.22 | 0.33 | 0.31 | 1.12 |
| Nainital | 2013 | 391.7 | 8.0 | 1963.1 | 58.2 | 223.2 | 69.7 | 549.9 | 24.8 | 327.2 | 1.03 | 8.11 | 0.15 | -8.19 | 0.39 | 0.47 | 1.29 |
| Nainital | 2014 | 394.3 | 7.5 | 1961.2 | 75.4 | 205.5 | 66.0 | 543.0 | 22.9 | 328.3 | 1.17 | 8.48 | 0.16 | -8.25 | 0.34 | 0.92 | 0.93 |
| Nainital | 2015 | 396.0 | 8.3 | 1984.1 | 72.8 | 226.6 | 77.1 | 549.3 | 28.1 | 329.4 | 1.02 | 8.84 | 0.23 | -8.24 | 0.38 | 1.04 | 0.87 |
| Nainital | 2016 | 400.8 | 8.2 | 1990.0 | 62.8 | 227.6 | 77.7 | 557.1 | 24.1 | 329.9 | 0.92 | 9.05 | 0.14 | -8.36 | 0.39 | 0.92 | 1.10 |
| Nainital | 2017 | 401.6 | 8.5 | 2012.1 | 83.8 | 229.0 | 77.8 | 555.9 | 26.3 | 331.0 | 1.24 | 9.43 | 0.16 | -8.28 | 0.41 | 0.90 | 1.08 |
| Nainital | 2018 | 404.3 | 7.8 | 2013.8 | 67.9 | 225.1 | 82.8 | 559.7 | 33.2 | 332.2 | 0.95 | 9.74 | 0.14 | -8.36 | 0.36 | 0.91 | 1.10 |
| Nainital | 2019 | 406.3 | 8.8 | 2021.3 | 64.1 | 232.4 | 84.3 | 556.8 | 29.5 | 332.7 | 1.08 | 10.10 | 0.13 | -8.36 | 0.40 | 0.81 | 1.19 |
| Nainital | 2020 | 407.4 | 6.7 | 2037.3 | 88.2 | 206.8 | 75.0 | 563.8 | 48.8 | 334.0 | 1.32 | 10.43 | 0.17 | -8.33 | 0.31 | 0.66 | 1.21 |
| Comilla | 2013 | 393.7 | 9.0 | 2214.6 | 291.6 | 294.7 | 168.8 | 607.7 | 69.3 | 328.4 | 2.29 | 8.12 | 0.18 | -8.41 | 0.38 | 0.42 | 0.95 |
| Comilla | 2014 | 395.4 | 10.8 | 2274.0 | 402.3 | 318.6 | 162.2 | 612.1 | 53.7 | 330.0 | 2.36 | 8.46 | 0.16 | -8.44 | 0.45 | 0.52 | 0.82 |
| Comilla | 2015 | 395.6 | 7.2 | 2272.4 | 250.6 | 293.8 | 118.4 | 596.0 | 32.6 | 330.5 | 1.87 | 8.78 | 0.13 | -8.34 | 0.32 | 0.44 | 0.87 |
| Comilla | 2016 | 402.4 | 8.1 | 2363.3 | 399.5 | 292.5 | 119.9 | 652.5 | 81.0 | 330.9 | 1.75 | 9.01 | 0.16 | -8.54 | 0.35 | 0.11 | 1.17 |
| Comilla | 2017 | 404.6 | 8.8 | 2484.5 | 450.1 | 293.4 | 129.2 | 601.9 | 27.6 | 332.1 | 2.29 | 9.37 | 0.19 | -8.54 | 0.38 | -0.14 | 1.23 |
| Comilla | 2018 | 403.8 | 8.1 | 2380.0 | 253.4 | 295.7 | 135.4 | 669.3 | 85.6 | 333.0 | 1.82 | 9.68 | 0.10 | -8.47 | 0.34 | 0.16 | 0.86 |
| Comilla | 2019 | 408.9 | 7.9 | 2406.7 | 331.5 | 284.5 | 114.0 | 604.6 | 36.9 | 333.9 | 1.81 | 10.07 | 0.16 | -8.58 | 0.33 | -0.06 | 1.44 |
| Comilla | 2020 | 415.2 | 11.2 | 2830.6 | 679.6 | 339.9 | 167.4 | 639.0 | 91.8 | 336.0 | 3.08 | 10.46 | 0.24 | -8.73 | 0.50 | -0.31 | 1.15 |






Table 2. Mean annual amplitudes of seasonal variation in atmospheric mole fractions of $CO_2$, $CH_4$, CO, $H_2$, $N_2O$, and $SF_6$
and $\delta^{13}C\text{-}CO_2$ and $\delta^{18}O\text{-}CO_2$ at Nainital (NTL) during 2007–2019 and at Comilla (CLA) during 2013–2019.

| Site | $CO_2$ | $CH_4$ | CO | $H_2$ | $N_2O$ | $SF_6$ | $\delta^{13}C\text{-}CO_2$ | $\delta^{18}O\text{-}CO_2$ |
|---|---|---|---|---|---|---|---|---|
| | ppm | ppb | ppb | ppb | ppb | ppt | ‰ | ‰ |
| Nainital | 22.1 ± 3.9 | 114 ± 52 | 153 ± 44 | 50.3 ± 18.0 | 1.01 ± 0.74 | 0.18 ± 0.16 | 0.96 ± 0.16 | 2.71 ± 0.79 |
| Comilla | 20.3 ± 5.7 | 486 ± 225 | 356 ± 90 | 70.4 ± 41.2 | 4.25 ± 1.45 | 0.23 ± 0.08 | 0.85 ± 0.19 | 2.33 ± 0.49 |







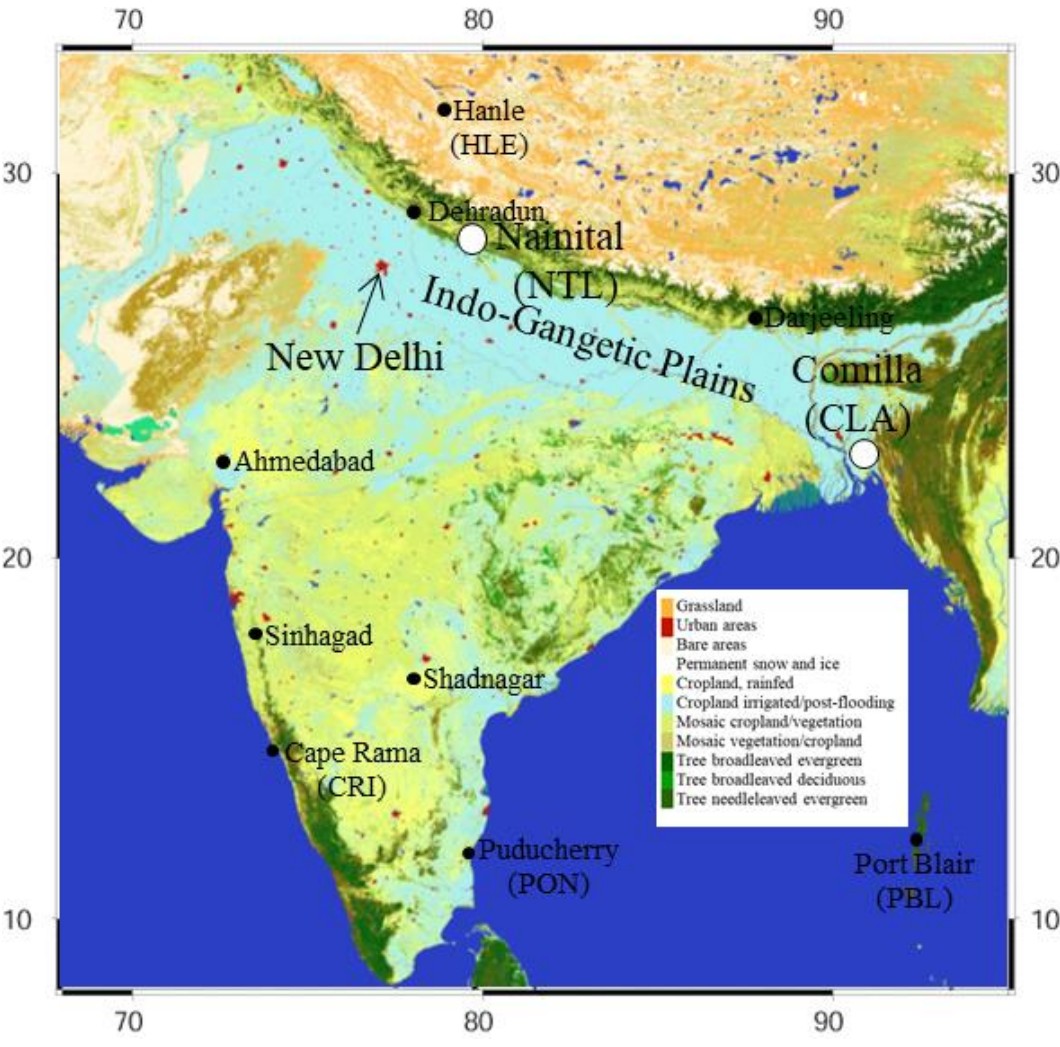


Figure 1. Location of Nainital (NTL), India (29.36°N, 79.46°E, 1940 m a.s.l.) and Comilla (CLA), Bangladesh (23.43°N,
91.18°E, 30 m a.s.l.) and other Indian sites for greenhouse gas (GHG) observation (Bhattacharya et al. 2009; Ganesan et al.
2013; Sharma et al., 2013; Tiwari et al., 2014; Lie et al., 2015; Sreenivas et al., 2016; Chandra et al.; 2016) and showing land
cover around the South Asia region (Arino et al., 2012).





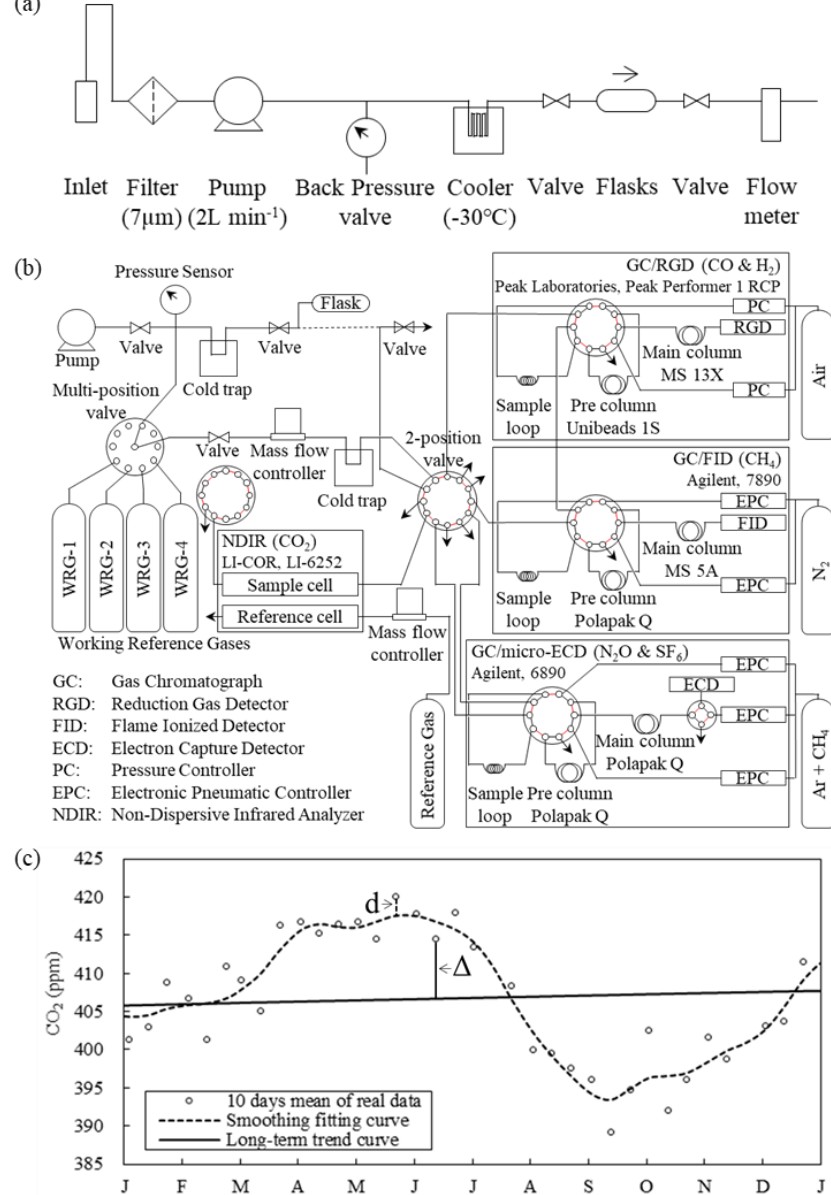

Figure 2. The line used for (a) flask sampling, (b) schematic of measurements of the dry-air mole fraction in the laboratory
and (c) diagram of the calculation method for "$\Delta$" term (e.g., $\Delta CO_2$) which was calculated by subtraction of the long-term
trend curve from 10 days mean of real data and "d" term (e.g., $dCO_2$) which was characterized by the deviation of 10 days
mean of real data from the smoothing fitting curve.




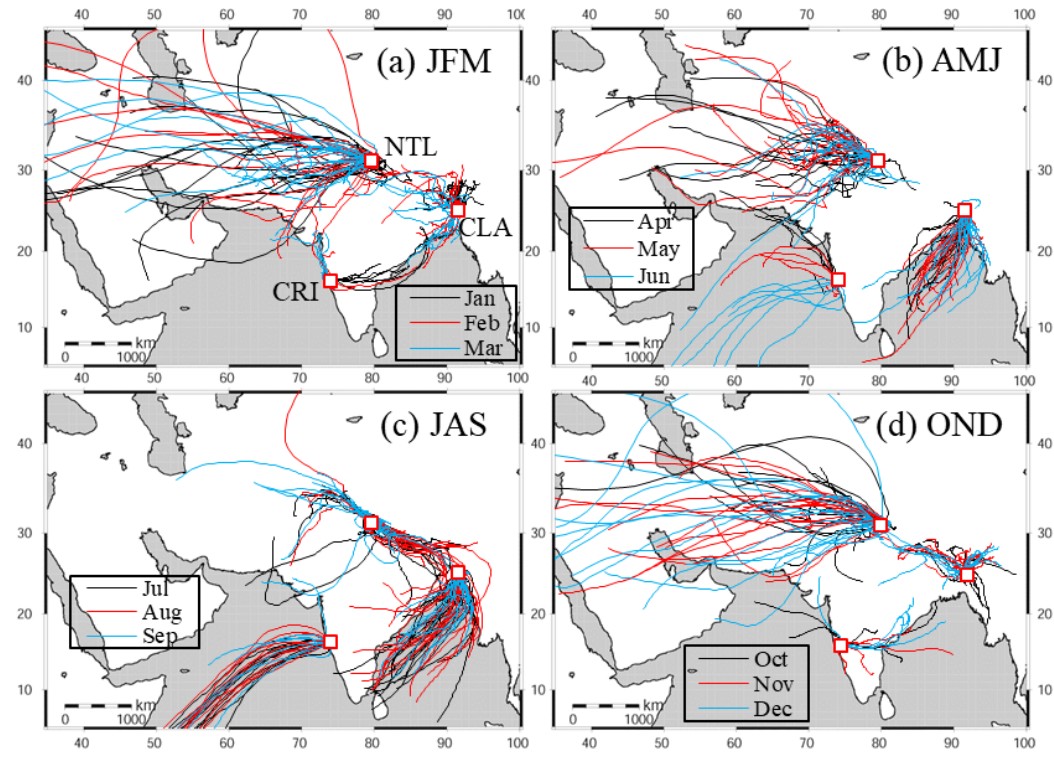


Figure 3. 72-hour back trajectory of Nainital (NTL), Comilla (CLA), and Cape Rama (CRI) in (a) January–March (JFM), (b) April–June (AMJ), (c) July–September (JAS), and (d) October–December. 72-hour back trajectory at NTL and CLA showed for 2012–2016 and the back trajectory at CRI showed for 2009–2013.






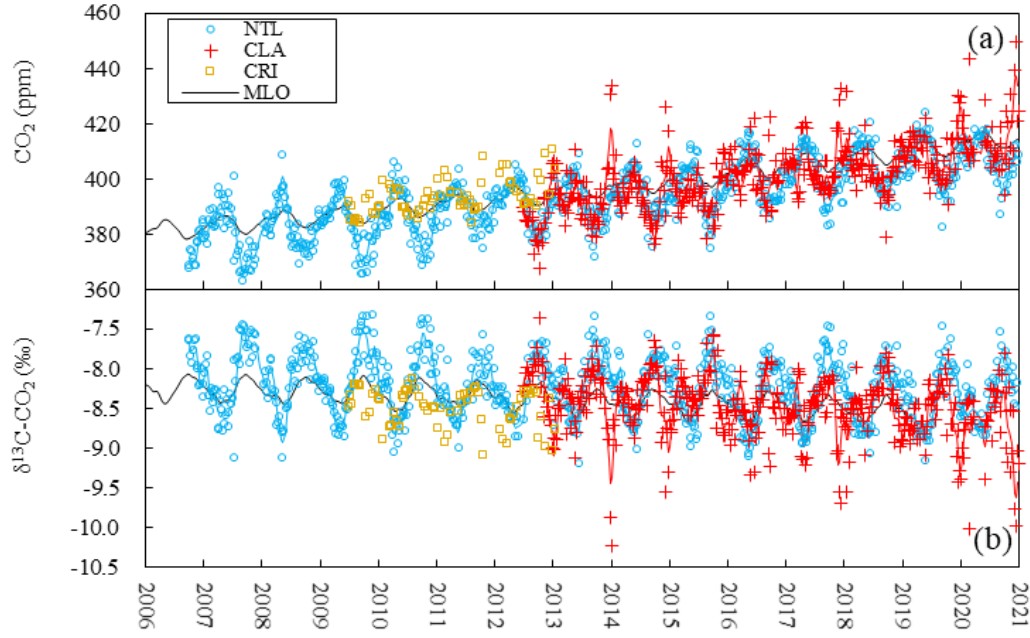

Figure 4. Time series of the (a) atmospheric $CO_2$ mole fraction, and (b) isotope ratio of $\delta^{13}C$-$CO_2$ at Nainital (NTL), Comilla
(CLA), Cape Rama (CRI), and Mauna Loa (MLO) in 2006–2020.



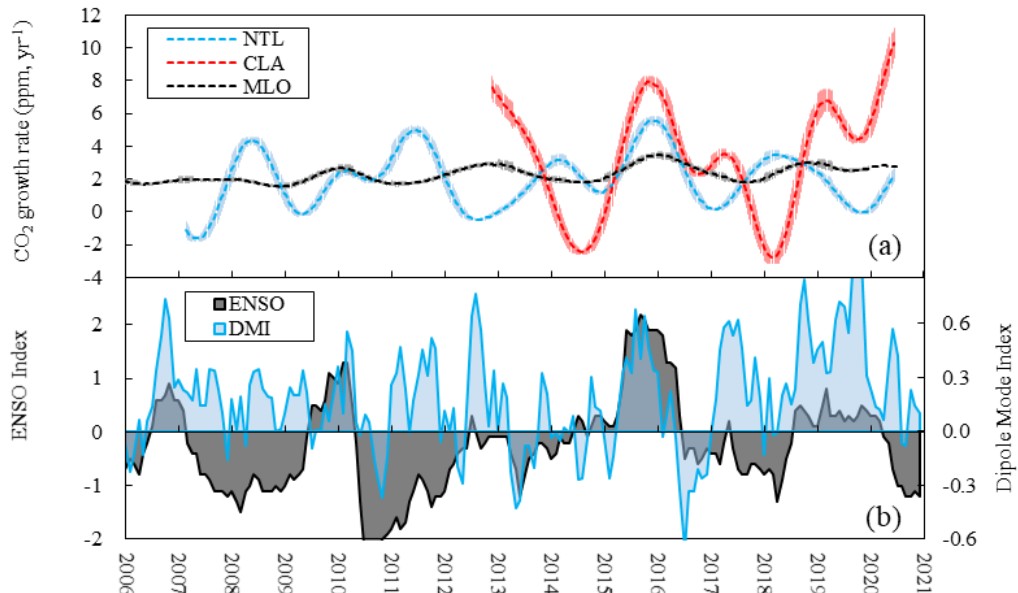

Figure 5. (a) Growth rates of the $CO_2$ mole fraction at Nainital (NTL), Comilla (CLA), and Mauna Loa (MLO) in 2006–
2020, and (b) the El Nino Southern Oscillation (ENSO) Index in 2006–2020 and the Dipole Mode Index (DMI) in 2006–

898    2020.






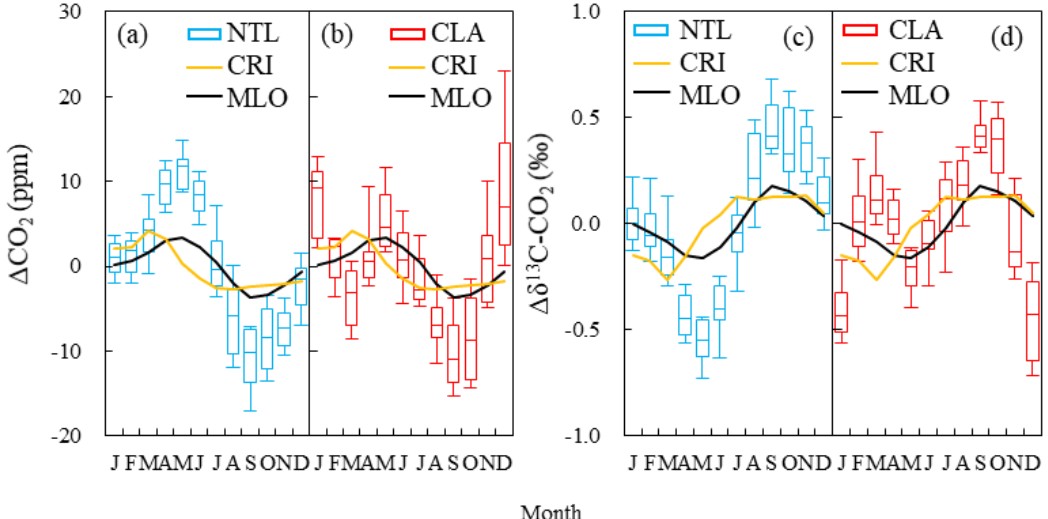


Figure 6. Seasonal variations in the $CO_2$ mole fraction at (a) Nainital (NTL) and (b) Comilla (CLA) and the isotope ratio of $\delta^{13}C$-$CO_2$ at (c) NTL and (d) CLA. Boxes with blue and red are for Nainital and Comilla and the black and yellow lines are for Mauna Loa (MLO) and Cape Rama (CRI), respectively. Median values (the line in the box), inner 50th percentile of the value (box), and inner 90th percentile of the value is from the monthly averaged $CO_2$ mole fractions.







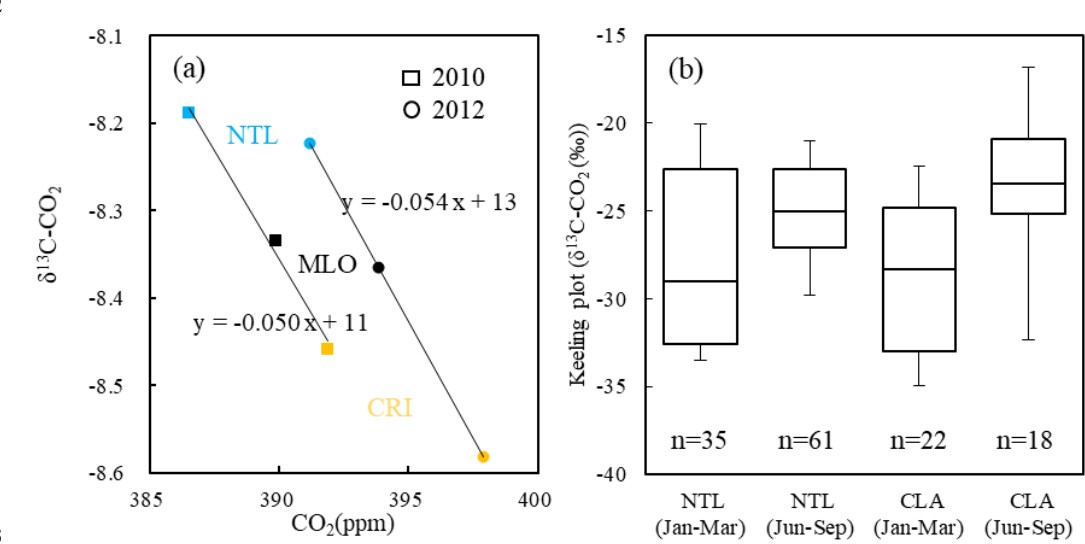


Figure 7. (a) Relationship between the annual values of the $CO_2$ mole fraction and isotopic ratio of $\delta^{13}C$-$CO_2$ at Nainital

(NTL), Cape Rama (CRI), and Mauna Loa (MLO) in 2010 and 2012, and (b) the intercept values of the Keeling plot of

Nainital and Comilla (CLA) in January–March and June–September.






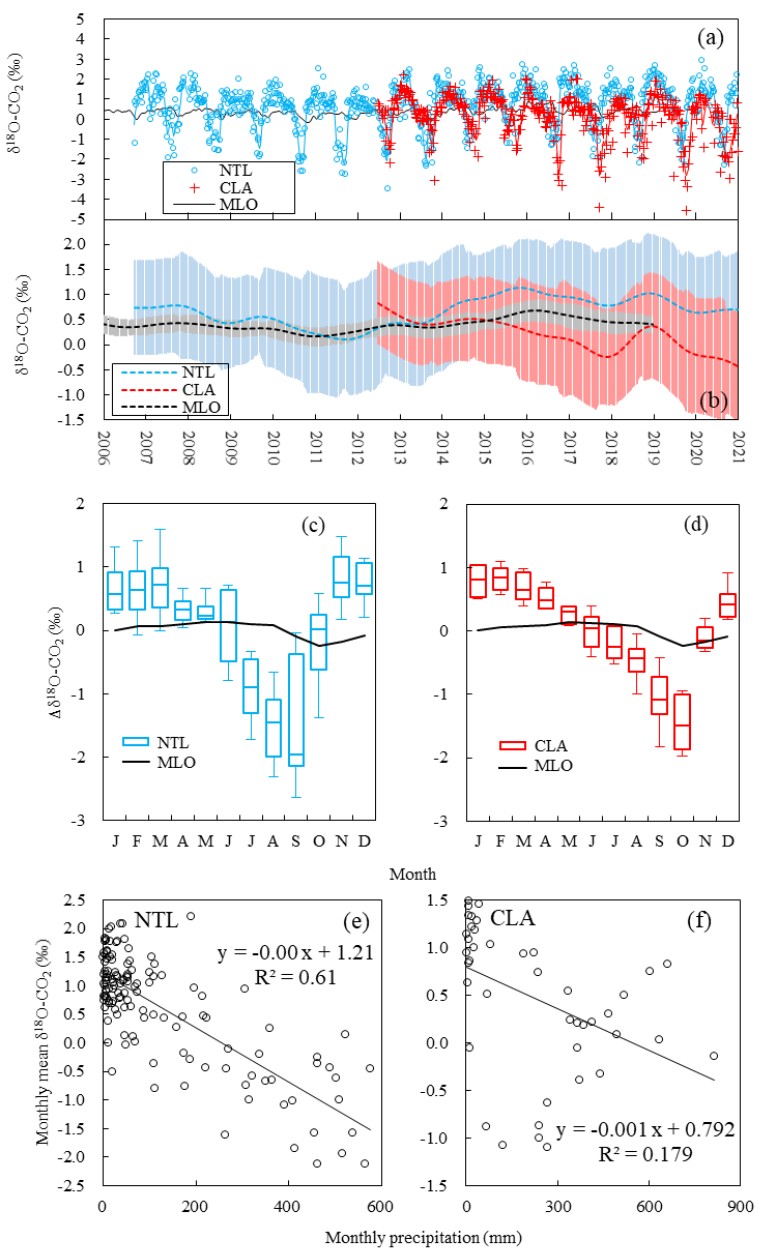



Figure 8. Time series of (a) measured values and (b) long-term trend for isotopic ratio of $\delta^{18}$O-CO$_2$ at Nainital (NTL),
Comilla (CLA), and Mauna Loa (MLO) in 2006–2020, the seasonal variation of $\delta^{18}$O-CO$_2$ at (c) NTL and (d) CLA, and the
relationship between monthly precipitation of the state of Uttarakhand and Bangladesh and the monthly mean of $\delta^{18}$O-CO$_2$ at
(e) NTL and (f) CLA.



Figure 9. Time series of (a) measured values and (b) growth rate of the CH$_4$ mole fraction at Nainital (NTL), Comilla (CLA),
Cape Rama (CRI), and Mauna Loa (MLO) in 2006–2020, the seasonal variation in the CH$_4$ mole fraction at (c) NTL and (d)
CLA, and the relationship between the short-term component of dCO and dCH$_4$ at (e) NTL and (f) CLA during January –
March (JFM), April–June (AMJ), July–September (JAS) and October–December (OND).


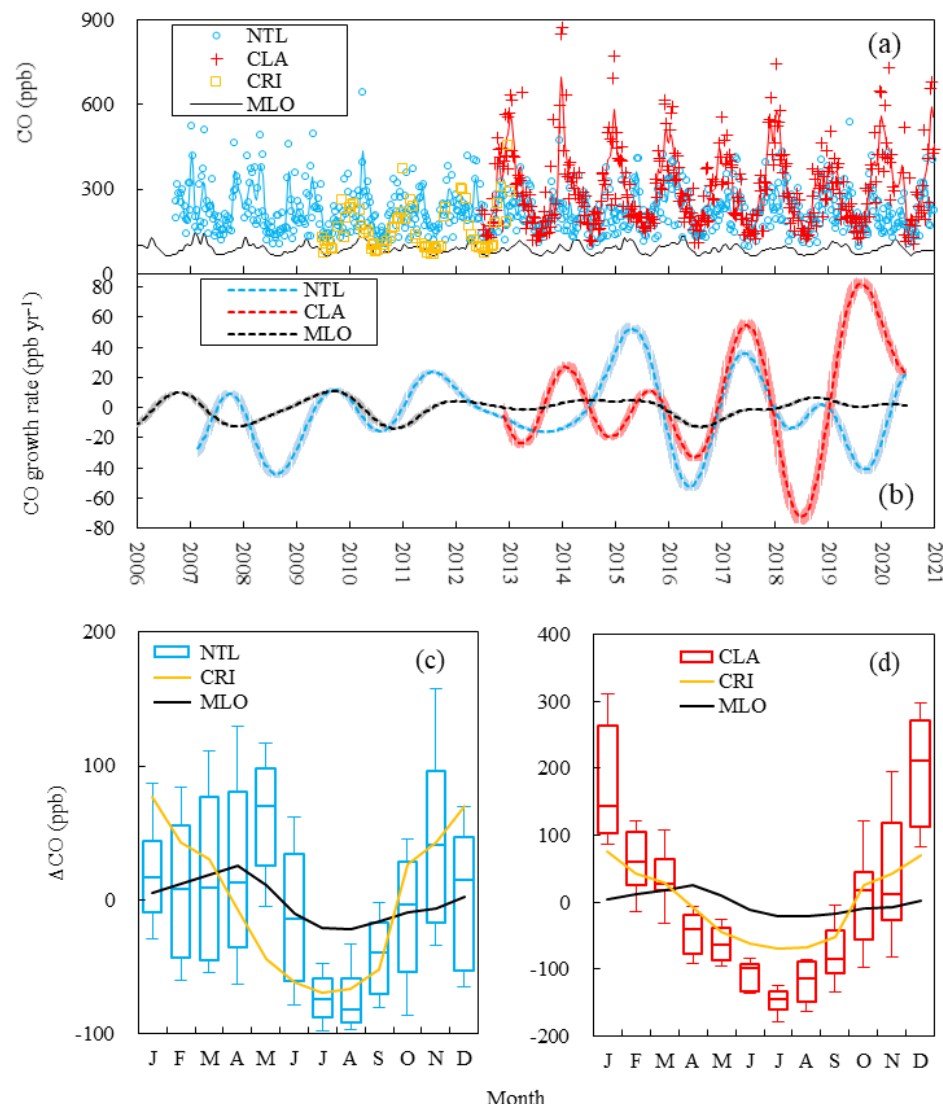


Figure 10. Time series of (a) measured values and (b) growth rates of CO mole fraction at Nainital (NTL), Comilla (CLA),
Cape Rama (CRI), and Mauna Loa (MLO) in 2006–2020, and the seasonal variation of CO mole fraction at (c) NTL and (d)
CLA.


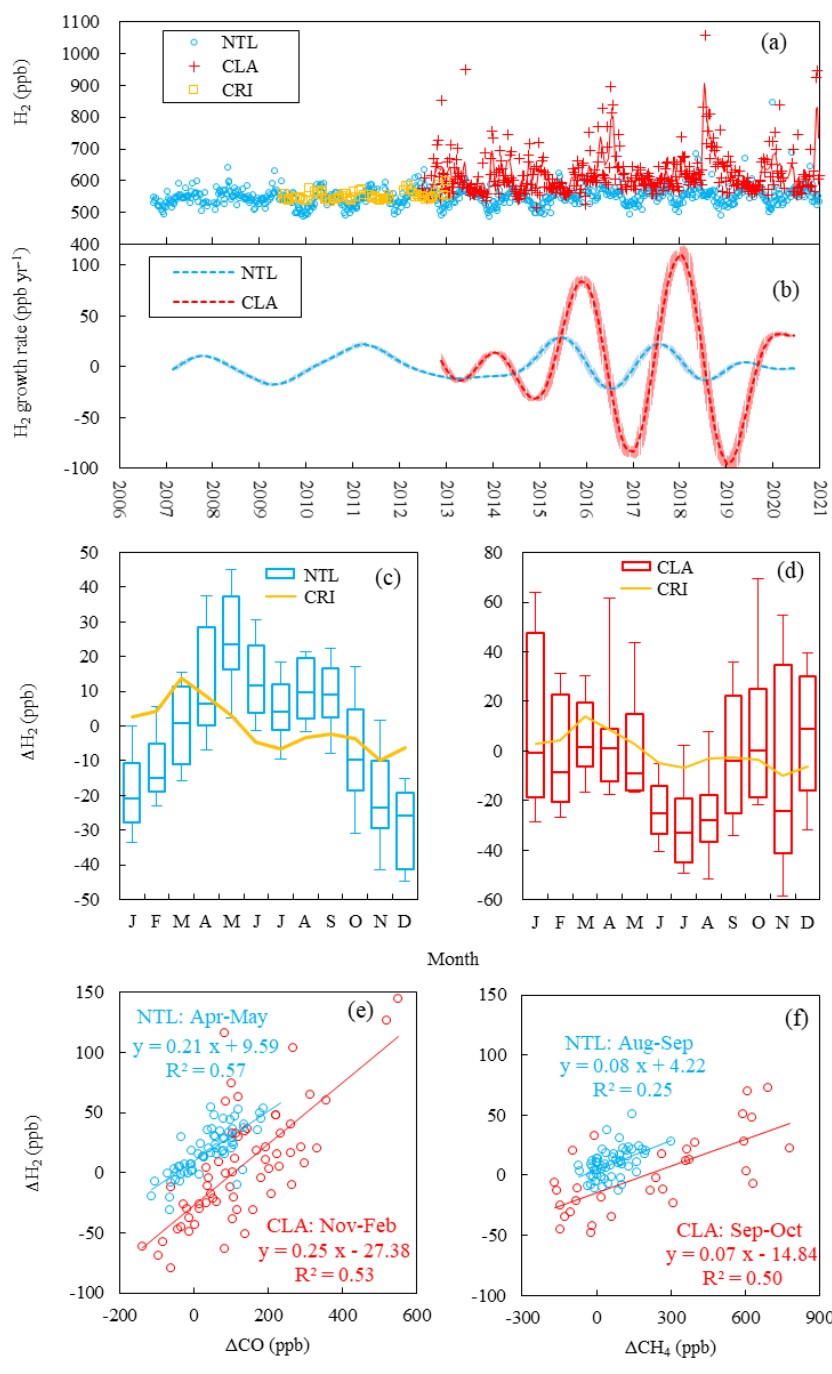



Figure 11. Time series of (a) measured values and (b) growth rate of the atmospheric $H_2$ mole fraction at Nainital (NTL),
Comilla (CLA), and Cape Rama (CRI) in 2006–2020, seasonal variation in the $H_2$ mole fraction at (c) NTL and (d) CLA,
and scatter plots for the relationship of (e) $\Delta H_2$ and $\Delta CO$ at NTL during April–May and at CLA during November–February
when biomass burning occurred frequently, and (f) $\Delta H_2$ and $\Delta CH_4$ at NTL during August–September and at CLA during
September–October when the maximum $CH_4$ mole fraction was measured.

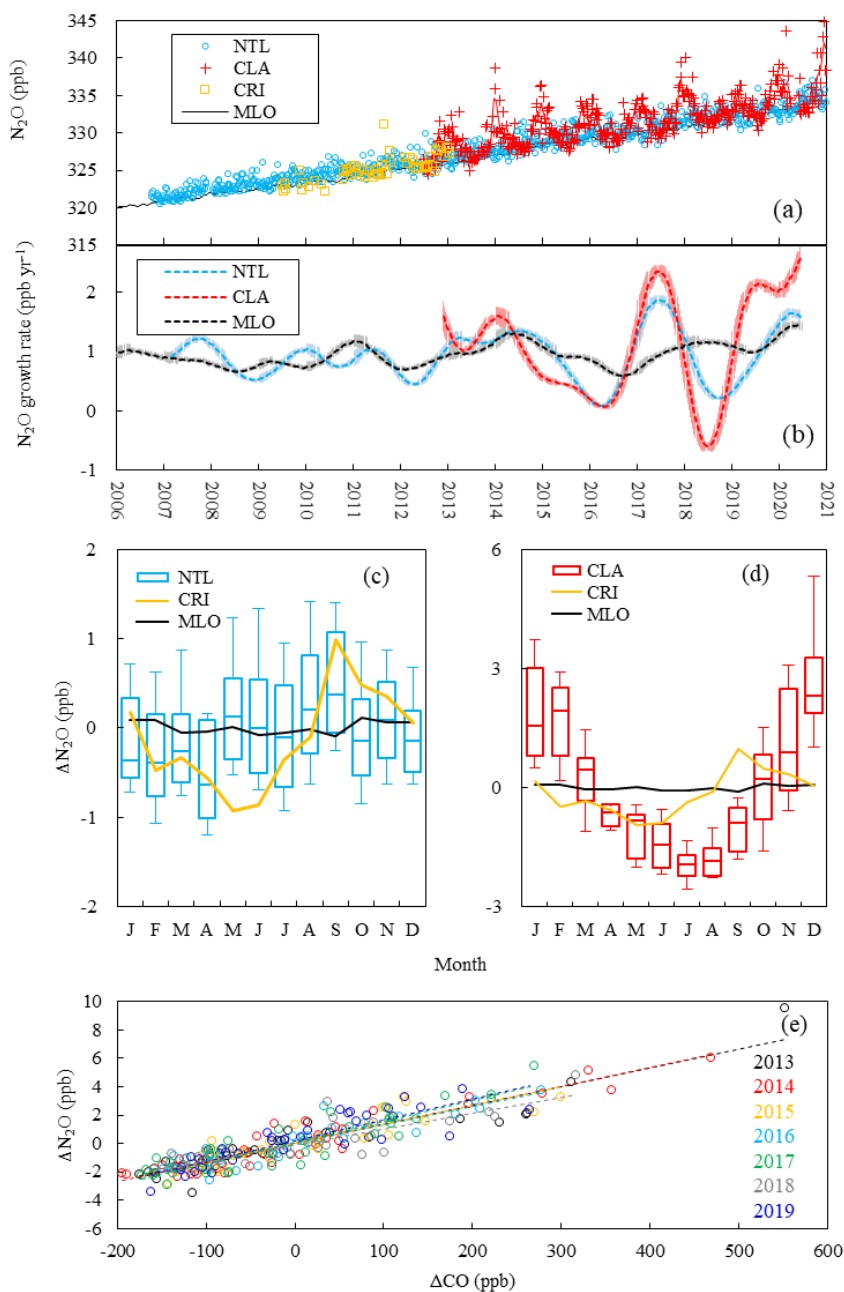


Figure 12. Time series of (a) measured values and (b) growth rates of the N₂O mole fraction at Nainital (NTL), Comilla (CLA),
and Mauna Loa (MLO) in 2006–2020, seasonal variations in the N₂O mole fraction at (c) NTL and (d) CLA, and (e) the
relationship between the ΔN₂O and ΔCO at CLA in 2013–2019.

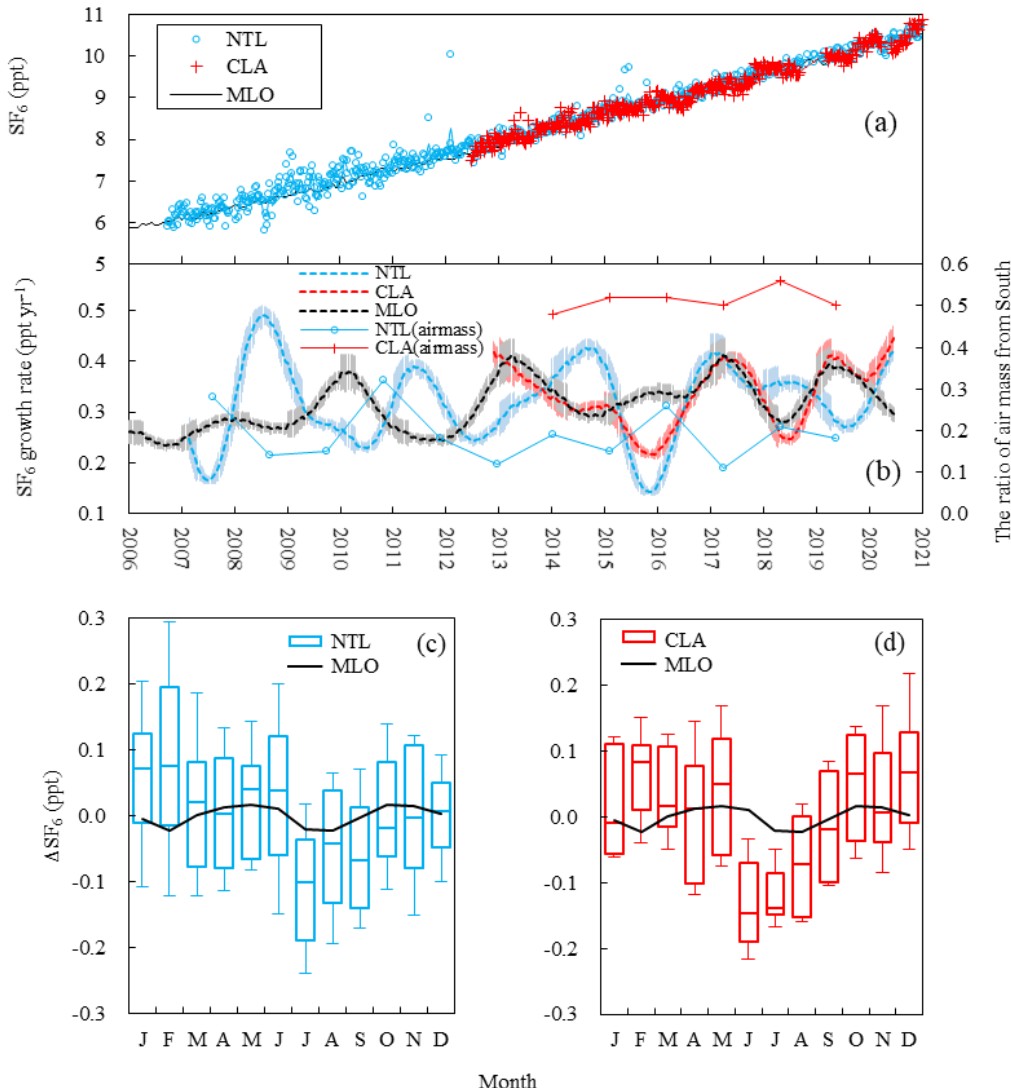

Figure 13. Time series of (a) measured values and (b) growth rates of the SF$_6$ mole fraction at Nainital (NTL), Comilla
(CLA), and Mauna Loa (MLO) and the ratios of the air mass from south at NTL and CLA in 2006–2020, and seasonal
variations in the SF$_6$ mole fraction at (c) NTL and (d) CLA.