# Peer review of "Measurement report: Regional characteristics of seasonal and long-term variations in greenhouse gases at Nainital, India and Comilla, Bangladesh"

_Atmospheric Chemistry and Physics, 2021_

## Referee Comment (RC2)

**Review of Nomura et al., 2021:** *Measurement report: Regional characteristics of seasonal and long-term variations in greenhouse gases at Nainital, India and Comilla, Bangladesh*

Nomura et al., present a new set of measurements from two sites on the Indian subcontinent, Nainital in northern India and Comilla in Bangladesh. Despite its large contribution to global greenhouse gas emissions, and the potential for future growth, atmospheric measurements from the region – required for top-down estimation of GHG emissions – are sparse. The analytical techniques described are appropriate, though I would like to see some additional information (see comments below), and the measurements themselves appear to be of high quality. For these reasons, the data presented merits publication in ACP, though I would like to see some consideration of the following points:

**Major comments:**
L103 – the authors state that they have estimated a small contribution from local sources. This might well be the case, but it would be good to know how this was estimated. The nearest populated region is fairly close for a background station, can the authors be sure that this urban area is not having a large effect on the measurements at NTL?

L112 – similar to my last comment, how can the authors be sure that CLA is not overwhelmingly influenced by local emissions. The inlet at CLA is fairly low (8 magl), and I would be concerned that local burning, agricultural emissions etc. might regularly 'drown out' regional signals. This requires some additional discussion in the main text.

L122 – given this is a 'measurement report', I would like to see some more detail on how the measurements were conducted. For instance, what was the procedure for analyzing $CO_2$ on the NDIR. Is the final measurement on average of a set-length injection? How often was the standard analyzed? It would also be good to see the average measurement precision for each species.

L169 – the back-trajectories shown are single particle trajectories. These trajectories don't appear to indicate when the particle is within close contact with the surface, and when it isn't. Without such information, the trajectories don't offer much additional information, e.g. a trajectory may originate over the Indo Gangetic Plain, but if the particle is many kilometers above the surface, it is unlikely to interact with potential sources? At the very least, this needs to be acknowledged in the main text.

L179 – I share the concerns of reviewer 1 with regards to the averaging of data into 10-day averages. It would seem to make more sense to calculate the long-term trends from the raw weekly data, as opposed to applying an average that in some cases only includes 1 data point. I would recommend calculating the long-term trends from the raw data or provide more detail on why a 10-day average is appropriate.

L280 onwards – I expect to see plenty of detail in a measurement report, but I found much of the results section to be overly verbose, to the point that it detracted from the main points of

discussion. I would suggest that the results section of the paper would benefit from some shortening, and that the authors concentrate on some of the more important findings. Specific examples:

- L263:300 – discussion of the different crop cycles is interesting, but does could be shortened and references condensed
- L454-482 – the conclusion that CO variability is linked to crop residue burning is compelling, however the same conclusion could be reached with significantly less text

**Technical corrections:**

L42-43 – end of first sentence needs restructuring

L43 – 'emerging' seems like a poor choice of word here. Perhaps 'developing' would be more appropriate

L82 – need to subscript CO2

L88 – 'believed to be'

L209 - 50–470 ppb of what?

L339 – typo 'fairy' needs to be corrected to fairly

L395 – the seasonality at Darjeeling is within the uncertainty of the seasonality estimated for CLA. Are the sources near to CLA similar to those at Darjeeling?

L492 – mainly should be 'main'

---

## Referee Comment (RC3)

Review comments:

Title: Measurement report: Regional characteristics of seasonal and long-term variations in greenhouse gases at Nainital, India and Comilla, Bangladesh
Author(s): Shohei Nomura et al.
MS No.: acp-2021-317

This study presents GHG observations over Northern Indian sites of Nainital NTL and Comilla CLA, Bangladesh. Factors like transported airmass, local cropping, biomass burning and precipitation locally seem to play a role in the observed variability at these sites. CLA show overall high CH4 concentration throughout the year. On the other hand, SF6 concentrations are similar to that at MLO, suggests that not many urban activities or anthropogenic emissions are active near these sites. This study emphasis that Indian Dipole DMI affects circulation and precipitation which in turn affects the growth rates of GHGs.

NTL and CLA long-term observations can play an important role towards understanding the regional carbon budget over South Asia. GHG variability in terms of seasonality, airmass transport dynamics, are already reported in various studies in the past (papers are cited in this study). However, studies reporting carbon flux estimation using top-down modelling are limited over this region.

Observations presented in this study are very useful to understand carbon budget over South Asia. NTL and CLA observed data should be available on public domain for other researchers at the earliest.

This manuscript may be accepted for publication in ACP after replying following comments.

1) L21-21: NTL do not show minima in Feb-March (ref. Fig. 6)
2) L25: "….in addition to other sources.." , what are other sources, pls specify.
3) L26-27: "High CH4 mole fractions……Plain", Is it due to large scale airmass transport or local emission?
4) L32-33: SF6 mole fraction is similar to that at MLO this suggests that there are few anthropogenic emissions sources near those places. However, CO observations are high at both the sites. Is it not a that a contradictory result?

5) L49-50: "…because there are few measured GHG mole fractions in the South Asian region" ; "Several observations on GHG mole fractions in the atmosphere have been done around India"…….Two contradictory statement. Consider revising.

6) L56-59: Do you mean CH4 and CO sources are co-located over these regions. Consider revising text in these lines.

7) L83-84: "Thus the GHG observations …………..long-term trend remain limited". This sentence is not clear. What do you men by long term trend remail limited? Consider revising.

8) L84-94: "In this work ……………..ENSO index". Why this study is important and how it fills gap areas left behind from past studies. Consider revising this paragraph.

9) L100: "…Mt. Mauna Peak….", is it Manora Peak? Pls check and correct.

10) L103-104: "We estimated that …….nearby", have you estimated or assumed? If you estimated then what is the basis for estimation? Same for assumption

11) L107-109: "Farmers in Comilla ………nearby emissions", this indicates that CLA is strongly influenced by the local anthropogenic emissions. On the other hand, based on SF6 observations you say that these sites are free from local emissions (ref. abstract). Its better to be consistent with the site characteristics described in the text. Also, be consistent in mentioning site name. Use either Comilla or CLA.

12) L112: "…..(on the roof of the second floor of the station) in NTL…….", What is height of the inlet head from the roof surface? What is height of the canopy close to the inlet head?

13) L113: What sealing material used in Pyrex flasks? Is it comparable to the boro 3.3 flasks (from Normag) and PCTFE sealing material used at MPI Jena.

14) L121: Air samples were cooled at -30 dc while sampling at NTL and CLA. Again, they are cooled at -80 dc before injecting to the analytical system at NIES. An explanation should given about this. Whether cooling twice (double dehumidification) have any scientific basis?

15) L122: Fig.2b should be simplified for the ease of readers. Put the direction of sample flow. Too many text inside the figure makes it complicated. Avoid writing text such as "Peak labs, Peak Performer, Agilent 7890, etc." inside this figure. It may be mentioned in the figure caption.

16) L126-127: "….GC-ECD or GC-micro-ECD", which one is used exactly?

17) L135-149: A figure may be shown similar like Fig.2b

18) L151-158: MLO is a reference site, however CRI does not represent a global/continental signal. It's a sub-regional site. Airmasses arrive at CRI are different than that of NTL and CLA. Hanle (HLE: French controlled site in India) or Seychelles (SEY) better represents large air masses in this region and can be considered as reference site. HLE represents northern hemisphere and SEY southern hemisphere. I suggest replacing CRI with HLE and SEY.

19) L176-177: Give more details about calculating the ratio.

20) L180: How you supplemented the value of missing period and any error in it. Describe in detail.

21) L191: Consider revising title of section 3.1, use of word "levels" may be misleading. May be replaced with concentration and low-concentration, high-concentration in the text.

22) Fig. 4: MLO curve clearly not visible after 2013 onward. Also, I do not understand the scientific reason behind using CRI data here. It is used because data is freely available at WDCGG? HLE would have been better background site like MLO. CRI is not advisable to use as a reference unless strong scientific motive is described.

23) L223-224: "….CRI site represents Southern Hemisphere during JJAS…", in that case you can use Seychelles (SEY) which is better representative of southern hemisphere.

24) L227-233: CRI represents large part of Indian land mass during Nov-March. Oct and and April are air mass transition months (seasonal change). So CRI site observations are good example of seasonal reversal of wind pattern. GHG mole fractions at CRI during JJAS represents oceanic air masses (pristine environment) and rest of the months it represents Indian land mass. Mole fractions representing Indian land mass may dominate in annual average. Such discussions should be written in the section 3.2.1.

25) L234-235: As mean growth rate (CO2) at NTL and CLA agrees with MLO, curves showing this should be added in Fig. 5a

26) L238-252: NTL and CLA CO2 growth and its relationship with ENSO and IOD are discussed in these paragraphs. However, no such statement is made in the conclusions section. Please add few lines about this in conclusion section as well. I am very surprise to see that there is no relationship between NTL and ENSO index. As ENSO is global phenomenon so its impact also is global particularly in GHG observations. India faces drought during most ElNino years and photosynthesis

activities are weak during this period and so CO2 enhancement occurs. I suggest authors to re-check your analysis in the case.

Manuscript may be accepted for publication after addressing above comments. And GHG observations data at NTL and CLA should be made available to the researchers for further useful research.

---

## Author Comment (AC1)

**Measurement report: Regional characteristics of seasonal and longterm variations in greenhouse gases at Nainital, India and Comilla, Bangladesh, by S. Nomura, M. Naja, M. K. Ahmed, H. Mukai, Y. Terao, T. Machida, M. Sasakawa, and P. K. Patra**

**Response to Reviewers**

We would like to thank the reviewers for providing comments and suggestions in our manuscript. We revised the manuscript based on the comments. Comments and questions from reviewers are reproduced here in black. Responses to reviewers are written in red.

Anonymous Referee #1

This manuscript presents important observation data for major GHGs from the northern Indian region. The weekly flask samples taken at a northern Indian station (Nainital, NTL) and a Bangladesh station (Comilla, CLA) for 2006-2012 were analyzed for the atmospheric concentrations of $CO_2$ (and d13C-$CO_2$, d18O-$CO_2$), $CH_4$, CO, $H_2$, $N_2O$ and $SF_6$. Authors discussed their seasonal variabilities considering regional climate conditions and contributions of regional sources and sinks. This study expanded the GHGs datasets for the Indian subcontinent, which is one of the most important regions in terms of the GHGs global budget, and thus provides new information about the regional characteristic features of major GHGs. This paper contains significant material and merits publication in Atmospheric Chemistry and Physics. The following comments will be considered for minor revision.

>Thank you very much for reading the manuscript. We appreciate your constructive comments and suggestions.

Specific comments:

Authors use 10-day average values to calculate a long-term trend and a smooth fit. And the seasonal variabilities were based on the deviation of a 10-day mean from the long-term trend curve. Were those 10-day means determined from a 10-day "moving" average? Can you explain why the 10-day means were used? Actually weekly raw data used for 10 day averaging are only one or two, and thus original data features might be misled due to this averaging of inconsistent number of data.

>The date intervals of the original data must be qual interval in order for our script for calculating only the "long-term trend" and "smooth fitting curve" (based on FFT). Basically, the date interval of the flask sampling is every 7 days. But irregularly, the date intervals are 6 days, 8 days or 14 days. The reason for setting 10-day means is to reduce missing data in intervals smoothing the original data and

to run our script. Also, we calculated the long-term trend and a smooth fitting curve from the data set as the date intervals of 7-days mean (The mean is put the dummy data during the missing periods), 20-days mean and 30-days mean and checked those values. For other evaluations such as scatter diagram and seasonal variation, we used individual data itself.

For Fig. 4, the atmospheric CO2 concentrations at NTL and CLA were compared with those from two background stations, and seasonal high values in August-October were explained by influence of air masses passing over the Indo-Gangetic plain. In addition, other noticeable features for CLA are ca. 20 ppm higher CO2 concentrations peaks shown above the smooth fit, and the corresponding lowest d13C-CO2 values, which periodically appeared at the beginning of each year. Air mass trajectory analysis for those data points and appropriate explanations for those distinctive values need to be added.

>We added the sentence of explanation for the distinctive values in L280-282. It is "small episodic peaks of the atmospheric CO2 mole fraction and isotopic ratio of d13C-CO2 of CLA at the beginning of each year was influenced by the biomass burning for heating in the close region, which is considered to be inland area from the site according to the air trajectory analysis".

For the CO2 growth rates in Fig. 5, the observations at the Cape Rama (CRI) station on the western coast of India can be compared with those for NTL and CLA because CRI represents the SH regional background site.

>The periods of the data set of NTL and CLA are from 2006 and 2012. But, the periods of CRI data set are in Feb 1993- Oct 2002 and Jul 2009- Jan 2013 (We got CRI data set from the WDCGG web site). The records in CRI data set from 2006 that we would like to compare were too short to calculate the growth rate. So, we didn't add the data of CO2 growth rate of CRI in Fig. 5.

Line 378-379: the long term trend of d18O-CO2 at CLA (Fig. 8b) seems to decrease, and authors suggested the amount effect of precipitation increase. But d18O-CO2 of CLA in Fig. 8f doesn't seem correlated well with precipitation amount.

>We added the monthly mean data of the precipitation at CLA until Jul 2021. The precipitation at CLA in trends to increase. Relationship between the monthly mean of $\delta$18O-CO2 and the monthly mean of precipitation of CLA appears weaker than that of NTL. However, if the monthly mean $\delta$18-CO$_2$ at CLA adds one or two months of time lag to the monthly mean of the precipitation, the correlation coefficient ($R^2$) between the monthly mean $\delta^{18}$O-CO2 at CLA and the monthly mean of precipitation is 0.4 to 0.5. This sentence added in L396-398 in the paper. We think that the monthly mean of $\delta^{18}$O-

CO$_2$ at CLA related with the monthly mean of precipitation, although the direct relationship between the δ$^{18}$O-CO$_2$ and precipitation in CLA seems weak.

Fig. 9a showed that the pollution signals of CH4 concentrations at CLA increased after 2018. The increases are more noticeable in 2019-2020. If there is any possibility of recent changes in rice field area, could the observed change in CH4 pollution concentrations be related with the increased rice cultivation in this region?

>There is no change in the rice field area and rice cultivation in Bangladesh. The amount of fertilizer application in the rice field area increased slightly. The increase of CH4 mole fraction at CLA in 2019-2020 might be influenced by the regional climate condition (e.g., increase of precipitation) and the enhancement of the global CH4 emission in 2020.

Line 137: move "by MT-252" (Air d13C-CO2 and d18O-CO2 were measured by MT-252 using….)

>We moved "by MT-252" as suggested.

---

## Author Comment (AC2)

**Measurement report: Regional characteristics of seasonal and longterm variations in greenhouse gases at Nainital, India and Comilla, Bangladesh, by S. Nomura, M. Naja, M. K. Ahmed, H. Mukai, Y. Terao, T. Machida, M. Sasakawa, and P. K. Patra**

**Response to Reviewers**

Anonymous Referee #3

Nomura et al., present a new set of measurements from two sites on the Indian subcontinent, Nainital in northern India and Comilla in Bangladesh. Despite its large contribution to global greenhouse gas emissions, and the potential for future growth, atmospheric measurements from the region – required for top-down estimation of GHG emissions – are sparse. The analytical techniques described are appropriate, though I would like to see some additional information (see comments below), and the measurements themselves appear to be of high quality. For these reasons, the data presented merits publication in ACP, though I would like to see some consideration of the following points:

>Thank you very much for reading the manuscript. We appreciate your constructive comments and suggestions.

Major comments: L103 – the authors state that they have estimated a small contribution from local sources. This might well be the case, but it would be good to know how this was estimated. The nearest populated region is fairly close for a background station, can the authors be sure that this urban area is not having a large effect on the measurements at NTL?

>NTL is located the edge of Himalaya Mountain and faced the Indo-Gangetic Plain. The wind at NTL blows always from Indo-Gangetic Plain and the wind speed of NTL is 3-10 m/sec. The seasonal variation of GHGs mole fraction at NTL are stable every year. Also, the value of SF6 mole fraction is stable and almost same level with MLO, despite that major source of SF6 exist in the city. The representative nearest populated cities for NTL are Nainital city and Haldwani city. Those are located about 2 km northwest and 20 km southeast of the NTL site, representatively. Nainital city is leeward site of NTL site for most of the year and the altitude of Haldwani is 1000-m lower than the altitude of NTL. Therefore, atmosphere of NTL might be partly influenced by the nearest populated areas, however, mainly influenced by relatively larger air mass over western Indo-Gangetic Plain in terms of high altitude of the sampling site, wind condition, seasonal variation patterns of all observed GHGs.

L112 – similar to my last comment, how can the authors be sure that CLA is not overwhelmingly influenced by local emissions. The inlet at CLA is fairly low (8 magl), and I would be concerned that

local burning, agricultural emissions etc. might regularly 'drown out' regional signals. This requires some additional discussion in the main text.

> If CLA is influenced strongly by the emission from the farmer houses and Comilla city, we could see frequently the episodic CO enhancement by the biomass burning. However, the enhancement of CLA is hardly happen. Also, the seasonal variation of CO mole fraction at CLA are stable every year. The land use of the central region of Bangladesh is almost uniformity (small farming village and large paddies field) in a sense and geographical features is a flat. Therefore, we could see typical seasonal variation in this region, if we can avoid very local emission. Wind blows always over CLA, even though the speed is relative slow as 2-5 m/sec. We concluded that the atmosphere of CLA was influenced mainly by the air mass of large rural area of the central Bangladesh region from such situation and the GHG data (the stable seasonal variation of GHGs and no episodic CO enhancement of CLA). We added some sentences to the text according to above context.

L122 – given this is a 'measurement report', I would like to see some more detail on how the measurements were conducted. For instance, what was the procedure for analyzing CO2 on the NDIR. Is the final measurement on average of a set-length injection? How often was the standard analyzed? It would also be good to see the average measurement precision for each species.

>We added the sentence of "Sample was injected to the analytical system three times per one flask and the working standard gases were analyzed after every two flasks." In L136-137.
We added the sentences of "The mole fractions of respective working standard gases are 379.00, 403.01, 423.84 and 441.10 ppm for CO2, 1681.50, 1852.12, 1998.83 and 2167.63 ppb for CH4, 59.84, 164.57, 267.33 and 373.54 ppb for CO, 401.40, 502,98, 610.49, 715.95 ppb for H2, 319.23, 326.91, 337.53 and 345.54 ppb for N2O and 4.65, 9.77, 14.53 and 19.08 ppt for SF6." And " Analytical precision for repetitive measurements is less than 0.03 ppm for CO2, 1.7 ppb for CH4, 0.3 ppb for CO, 3.1 ppb for H2, 0.3 ppb for N2O, and 0.3 ppt for SF6 (Machida et al., 2008)." in L144-148.

L169 – the back-trajectories shown are single particle trajectories. These trajectories don't appear to indicate when the particle is within close contact with the surface, and when it isn't. Without such information, the trajectories don't offer much additional information, e.g. a trajectory may originate over the Indo Gangetic Plain, but if the particle is many kilometers above the surface, it is unlikely to interact with potential sources? At the very least, this needs to be acknowledged in the main text.

>We calculated altitude with latitude and longitude on the back trajectory analysis and we checked that the air mass at NTL and CLA passed through the atmospheric boundary layer from the data of

altitude. We added the sentence of "We referred the altitude data when we evaluated the effects of GHGs emissions sources near the surface." in L190-191.

L179 – I share the concerns of reviewer 1 with regards to the averaging of data into 10-day averages. It would seem to make more sense to calculate the long-term trends from the raw weekly data, as opposed to applying an average that in some cases only includes 1 data point. I would recommend calculating the long-term trends from the raw data or provide more detail on why a 10-day average is appropriate.

>The date intervals of the original data must be qual interval in order for our script for calculating only the "long-term trend" and "smooth fitting curve" (based on FFT). Basically, the date interval of the flask sampling is every 7 days. But irregularly, the date intervals are 6 days, 8 days or 14 days. The reason for setting 10-day means is to reduce missing data in intervals smoothing the original data and to run our script. Also, we calculated the long-term trend and a smooth fitting curve from the data set as the date intervals of 7-days mean (The mean is put the dummy data during the missing periods), 20-days mean and 30-days mean and checked those values. For other evaluations such as scatter diagram and seasonal variation, we used individual data itself. We added one sentence for explanation in the section.

L280 onwards – I expect to see plenty of detail in a measurement report, but I found much of the results section to be overly verbose, to the point that it detracted from the main points of discussion. I would suggest that the results section of the paper would benefit from some shortening, and that the authors concentrate on some of the more important findings. Specific examples:

>We removed several sentences to concentrate our discussion.

• L263:300 – discussion of the different crop cycles is interesting, but does could be shortened and references condensed

>We removed below sentence
"Especially, the $CO_2$ mole fraction at CLA in February–March decreased remarkably, by up to approximately 8 ppm."

"In the region near NTL, rice, wheat, and other cereals and millets were mainly cultivated (DAC/MA, 2015; SID/MP, 2018; and DES/MAFW, 2019)."

"Panigrahy et al. (2010) reported the main rice growing seasons in North India to be July–September and February–March by using the Normalized Difference Vegetation Index (NDVI). Nayak et al. (2010) also reported that Net Primary Productivity (NPP) on the Indo-Gangetic Plain increased in August–September and February–March, estimated from the NDVI".

• L454-482 – the conclusion that CO variability is linked to crop residue burning is compelling, however the same conclusion could be reached with significantly less text

>We removed the sentence of "(i.e., two mole fraction peaks in May and November)" and "Sharma et al. (2010) suggested that the high CO mole fraction on the Western Indo-Gangetic Plain is emitted in October by the burning of harvest residues, based on data from satellite observations.".

Technical corrections:
L42-43 – end of first sentence needs restructuring

>We changed "The atmospheric mole fractions of $CO_2$, $CH_4$, $N_2O$ and many other greenhouse gases (GHGs)" to "The mole fraction of many greenhouse gases (GHGs) in the atmosphere, including CO2, CH4, and N2O, has been increasing worldwide in recent years." in L43-44.

L43 – 'emerging' seems like a poor choice of word here. Perhaps 'developing' would be more appropriate

>We changed "developing" in L44 as suggested.

L82 – need to subscript CO2 L88 – 'believed to be' L209 - 50–470 ppb of what? L339 – typo 'fairy' needs to be corrected to fairly

>We changed "$CO_2$" as suggested.
>We changed "believed to be" in L91 as suggested.
>We changed "50-470 ppb for $CH_4$" in L227 as suggested.
>We changed "fairly" in L354 as suggested.

L395 – the seasonality at Darjeeling is within the uncertainty of the seasonality estimated for CLA. Are the sources near to CLA similar to those at Darjeeling?

>Darjeeling is affected by the flesh air mass from the South Hemisphere in the monsoon season and the air mass with the high CH4 concentration from the paddies field in Indo-Gangetic Plain in the non-monsoon season like CLA. But, CH4 mole fraction at Darjeeling is lower than that of CLA because CLA are located at t the central area in vast paddies field region.

L492 – mainly should be 'main'

>We changed "main" in L 508 as suggested.

---

## Author Comment (AC3)

**Measurement report: Regional characteristics of seasonal and longterm variations in greenhouse gases at Nainital, India and Comilla, Bangladesh, by S. Nomura, M. Naja, M. K. Ahmed, H. Mukai, Y. Terao, T. Machida, M. Sasakawa, and P. K. Patra**

**Response to Reviewers**

We would like to thank the reviewers for providing comments and suggestions in our manuscript. We revised the manuscript based on the comments. Comments and questions from reviewers are reproduced here in black. Responses to reviewers are written in red.

Anonymous Referee #4

This study presents GHG observations over Northern Indian sites of Nainital NTL and Comilla CLA, Bangladesh. Factors like transported airmass, local cropping, biomass burning and precipitation locally seem to play a role in the observed variability at these sites. CLA show overall high CH4 concentration throughout the year. On the other hand, SF6 concentrations are similar to that at MLO, suggests that not many urban activities or anthropogenic emissions are active near these sites. This study emphasis that Indian Dipole DMI affects circulation and precipitation which in turn affects the growth rates of GHGs.

NTL and CLA long-term observations can play an important role towards understanding the regional carbon budget over South Asia. GHG variability in terms of seasonality, airmass transport dynamics, are already reported in various studies in the past (papers are cited in this study). However, studies reporting carbon flux estimation using top-down modelling are limited over this region.

Observations presented in this study are very useful to understand carbon budget over South Asia. NTL and CLA observed data should be available on public domain for other researchers at the earliest. This manuscript may be accepted for publication in ACP after replying following comments.

>Thank you very much for seeing the value of our observations. We appreciate your constructive comments and suggestions.

1) L21-21: NTL do not show minima in Feb-March (ref. Fig. 6)

>We removed "NTL" in L21 and added the sentence in L22-23 as follows: "Although NTL had only one clear minima in September,…".

2) L25: "….in addition to other sources.." , what are other sources, pls specify.

>Biomass burning would be partly contributed to high $CH_4$ in August–October but we need further studies for estimates of contributions from biomass burnings. We removed "in addition to other sources" and modified the sentence as "mainly due to the influence of $CH_4$ emissions from the paddy fields." in L25-26.

3) L26-27: "High CH4 mole fractions……Plain", Is it due to large scale airmass transport or local emission?

>High $CH_4$ mole fractions were affected by the both local emission and air mass transport over Indo-Gangetic Plain. We added the sentence: "which were affected by the both local emission and air mass transport." in L27.

4) L32-33: SF6 mole fraction is similar to that at MLO this suggests that there are few anthropogenic emissions sources near those places. However, CO observations are high at both the sites. Is it not a that a contradictory result?

>$SF_6$ is used mainly for high voltage equipment such as step-up or down transformer and electrical plants. Some portion will be come from the process of its production. But CO is emitted mainly from the biomass burning. Thus, $SF_6$ emission source is completely different from the CO emission source. We modified the sentence of "there were few anthropogenic emission sources" to "there were few anthropogenic $SF_6$ emission sources" in L33.

5) L49-50: "…because there are few measured GHG mole fractions in the South Asian region" ; "Several observations on GHG mole fractions in the atmosphere have been done around India"…….Two contradictory statement. Consider revising.

>We changed the expression of "few" to "only a few". We removed the sentence of "Several observations on GHG mole fractions in the atmosphere have been done around India" in L50.

6) L56-59: Do you mean CH4 and CO sources are co-located over these regions. Consider revising text in these lines.

>This paper didn't indicate that CH4 and CO sources are co-located over these regions. This paper just indicates that Indian subcontinent has strong emission sources of CH4 and CO and the atmospheric mole fractions of CH4 and CO is affected by the seasonal wind. Major CH4 emission source and major CO emission source is different from the results of our flask sampling. Major CH4 emission source is

the paddies field and the major CO emission source is the biomass burning.

7) L83-84: "Thus the GHG observations …………..long-term trend remain limited". This sentence is not clear. What do you men by long term trend remail limited? Consider revising.

>We modified the sentence: "Thus, the GHGs observation program in Indian region is expanding gradually, however, the characterization of GHGs behaviour in the northern Indian subcontinent and their long-term trends are not well understood.". in L83.

8) L84-94: "In this work ……………..ENSO index". Why this study is important and how it fills gap areas left behind from past studies. Consider revising this paragraph.

>We added the sentence of "In this paper, we present the longer GHGs data than previous studies in the Indo-Gangetic Plain including Bangladesh, which is a blank area for GHGs observation and clarify the characteristics of GHGs in the Indian subcontinent by analyzing the periodicity of GHGs growth rates and comparing them with regional climatic conditions." in L84-87.

9) L100: "…Mt. Mauna Peak….", is it Manora Peak? Pls check and correct.

>We modified "Mt. Mauna Peak" to Mt. Manora Peak" in L103 as your suggestion.

10) L103-104: "We estimated that …….nearby", have you estimated or assumed? If you estimated then what is the basis for estimation? Same for assumption

>We modified "which mean that NTL might be influenced mainly by the air mass passing through the Indo-Gangetic Plain. We estimated that the air of NTL is not strongly influenced by local GHGs emissions nearby." to "which mean that the air of NTL is influenced mainly by the air mass passing through the Indo-Gangetic-Plain, rather than extremely influenced by local GHGs emissions nearby." in L105-107.

11) L107-109: "Farmers in Comilla ………nearby emissions", this indicates that CLA is strongly influenced by the local anthropogenic emissions. On the other hand, based on SF6 observations you say that these sites are free from local emissions (ref. abstract). Its better to be consistent with the site characteristics described in the text. Also, be consistent in mentioning site name. Use either Comilla or CLA.

>SF$_6$ is not emitted by the biomass burning by the farmer. We modified "Comilla" to "CLA" in L109.

12) L112: "…..(on the roof of the second floor of the station) in NTL…….", What is height of the inlet head from the roof surface? What is height of the canopy close to the inlet head?

>Height of the inlet is 1-m higher than the roof surface. We added the information of the height of the inlet of NTL to the canopy in L117-118.

13) L113: What sealing material used in Pyrex flasks? Is it comparable to the boro 3.3 flasks (from Normag) and PCTFE sealing material used at MPI Jena.

>We use the Viton O-rings for sealing. We added the information of sealing in L 122.

14) L121: Air samples were cooled at -30 dc while sampling at NTL and CLA. Again, they are cooled at -80 dc before injecting to the analytical system at NIES. An explanation should given about this. Whether cooling twice (double dehumidification) have any scientific basis?

>To dry air samples almost completely before analysis is essential, because CO$_2$ and other GHGs should be measured on dry air base. So, we use -80dc coolant basically for our analytical system. But for sampling, we just use rather simple cooling system to prevent water from condensing on the inner surface of the glass flask.

15) L122: Fig.2b should be simplified for the ease of readers. Put the direction of sample flow. Too many text inside the figure makes it complicated. Avoid writing text such as "Peak labs, Peak Performer, Agilent 7890, etc." inside this figure. It may be mentioned in the figure caption.

>We added the sample flow to the Fig.2b.
We removed "Peak labs, Peak Performer, Agilent 7890, etc" in the Fig.2b.

16) L126-127: "….GC-ECD or GC-micro-ECD", which one is used exactly?

>We modified the sentence "a gas chromatograph with an electron capture detector or a micro electron capture detector (GC-ECD or GC-micro-ECD" to "a gas chromatograph with an electron capture detector (GC-ECD) until 2011 and with a micro electron capture detector (GC-micro-ECD) from 2012" in L134-135.

17) L135-149: A figure may be shown similar like Fig.2b

>The analysis line of GHGs mole fraction in the NIES laboratory is complex and consists of multiple instruments. We added the schematic because it contributes to understand the analysis line of GHGs mole fraction in the NIES laboratory. While, the analysis line of isotopic ratio in the NIES laboratory is simple and consists of one instrument. So, we judged that the schematic of the analysis line of isotopic ratio in the NIES laboratory isn't necessary in this paper.

18) L151-158: MLO is a reference site, however CRI does not represent a global/continental signal. It's a sub-regional site. Airmasses arrive at CRI are different than that of NTL and CLA. Hanle (HLE: French controlled site in India) or Seychelles (SEY) better represents large air masses in this region and can be considered as reference site. HLE represents northern hemisphere and SEY southern hemisphere. I suggest replacing CRI with HLE and SEY.

>We selected MLO and CRI for comparing with NTL and CLA because the data of MLO has representative of middle latitude of northern hemisphere and the data of CRI includes the characteristic of GHGs of India subcontinent. The mole fraction of GHGs in CRI shows the same level with the data of MLO when the air mass transported from Indian Ocean, while the mole fraction of GHGs in CRI shows the high concentration when the air mass transported over the India subcontinent. We didn't select SEY and HLN for comparing with NTL and CLA because the data of SEY were similar to the data of MLO and the SEY was located in southern hemisphere. We didn't use HLE because it is located at north area of Himalaya Mountain: We need to select the comparison site in south area from the Himalaya Mountain because the north area and the south area of Himalaya Mountain are quite different about the transportation of air mass, terrestrial condition and anthropogenic activities.

19) L176-177: Give more details about calculating the ratio.

>We modified "the ratio of air mass from south was calculated by the frequency of the air mass from south side on the flask sampling date with reference to the backward air trajectories data." to "the ratio of air mass from south per year was calculated by the frequency of the air mass from south side of Indian Ocean on the flask sampling date in each year with reference to the backward air trajectories data calculated by METEX." in L192-193.

20) L180: How you supplemented the value of missing period and any error in it. Describe in detail.

>The value of the missing period was supplemented with an interpolated values from the previous and

following data of the missing period for calculating the continuous long-term trend and smoothing fitting curve. We described in the text.

21) L191: Consider revising title of section 3.1, use of word "levels" may be misleading. May be replaced with concentration and low-concentration, high-concentration in the text.

>We replaced "level" to "mole fraction" or its "values".

22) Fig. 4: MLO curve clearly not visible after 2013 onward. Also, I do not understand the scientific reason behind using CRI data here. It is used because data is freely available at WDCGG? HLE would have been better background site like MLO. CRI is not advisable to use as a reference unless strong scientific motive is described.

>The mole fraction of GHGs in CRI shows the same level with the data of MLO when the air mass transported from Indian Ocean, while the mole fraction of GHGs in CRI shows the high concentration when the air mass transported over the India subcontinent. The data of CRI are very helpful in characterizing the behavior of GHGs mole fraction in India subcontinent. The data of HLE doesn't clear the seasonal variation of GHGs by the monsoon in India subcontinent. Also, we got the data of CRI on the WDCGG web site with contact to CSIRO staff.

23) L223-224: "….CRI site represents Southern Hemisphere during JJAS…", in that case you can use Seychelles (SEY) which is better representative of southern hemisphere.

>It is important that CRI is located in India subcontinent. We wanted to indicate that the data of CRI is influenced by the air mass from southern hemisphere during monsoon season only and same level of mole fraction with the data of MLO (also SEY) during monsoon season.

24) L227-233: CRI represents large part of Indian land mass during Nov-March. Oct and April are air mass transition months (seasonal change). So CRI site observations are good example of seasonal reversal of wind pattern. GHG mole fractions at CRI during JJAS represents oceanic air masses (pristine environment) and rest of the months it represents Indian land mass. Mole fractions representing Indian land mass may dominate in annual average. Such discussions should be written in the section 3.2.1.

>The same content is written in the first paragraph in the section.

25) L234-235: As mean growth rate (CO2) at NTL and CLA agrees with MLO, curves showing this should be added in Fig. 5a

>Thank you for your suggestion. We added the values of annual mean growth rate (CO2) at NTL and CLA in Table 1.

26) L238-252: NTL and CLA CO2 growth and its relationship with ENSO and IOD are discussed in these paragraphs. However, no such statement is made in the conclusions section. Please add few lines about this in conclusion section as well. I am very surprise to see that there is no relationship between NTL and ENSO index. As ENSO is global phenomenon so its impact also is global particularly in GHG observations. India faces drought during most ElNino years and photosynthesis activities are weak during this period and so CO2 enhancement occurs. I suggest authors to re-check your analysis in the case.

>We added "Indian Ocean Dipole" in L647 and L650. We checked the relationship of CO2 growth rate of Indian subcontinent sites and ENSO index many times because we also were surprise to see that in first time. We re-check the relationships after we got the CO2 data record in 30-50 years of Indian continent sites. As a side note, CO2 growth rate of few sites in Inland of Eurasia (Kazakhstan, Russia and China) also have anticorrelation with ENSO index.

Manuscript may be accepted for publication after addressing above comments. And GHG observations data at NTL and CLA should be made available to the researchers for further useful research.

>We are ready to open our observational data soon after publication.

Revised by authors:

---

## Referee Report (RR1)

Review comments: (Revised manuscript, acp-2021-317)

Title: Measurement report: Regional characteristics of seasonal and long-term variations in greenhouse gases at Nainital, India and Comilla, Bangladesh
Author(s): Shohei Nomura et al.
MS No.: acp-2021-317

Decision:
Authors have revised the manuscript extensively and addressed all the comments. Manuscript may be accepted in current form.